# Translation efficiency driven by CNOT3 subunit of the CCR4-NOT complex promotes leukemogenesis

Maryam Ghashghaei[1,2,12], Yilin Liu[2,3,12], James Ettles[4,5], Giuseppe Bombaci [6], Niveditha Ramkumar[2], Zongmin Liu[1,2], Leo Escano[2], Sandra Spencer Miko [7], Yerin Kim[2,8], Joseph A. Waldron[4,5], Kim Do[9], Kyle MacPherson[2], Katie A. Yuen[2], Thilelli Taibi[2], Marty Yue[2], Aaremish Arsalan[2], Zhen Jin[1,2], Glenn Edin[2], Aly Karsan [7], Gregg B. Morin [7,10], Florian Kuchenbauer [2], Fabiana Perna [6,11], Martin Bushell[4,5] & Ly P. Vu [1,2] ✉

Protein synthesis is frequently deregulated during tumorigenesis. However, the precise contexts of selective translational control and the regulators of such mechanisms in cancer is poorly understood. Here, we uncovered CNOT3, a subunit of the CCR4-NOT complex, as an essential modulator of translation in myeloid leukemia. Elevated CNOT3 expression correlates with unfavorable outcomes in patients with acute myeloid leukemia (AML). CNOT3 depletion induces differentiation and apoptosis and delayed leukemogenesis. Transcriptomic and proteomic profiling uncovers c-MYC as a critical downstream target which is translationally regulated by CNOT3. Global analysis of mRNA features demonstrates that CNOT3 selectively influences expression of target genes in a codon usage dependent manner. Furthermore, CNOT3 associates with the protein network largely consisting of ribosomal proteins and translation elongation factors in leukemia cells. Overall, our work elicits the direct requirement for translation efficiency in tumorigenesis and propose targeting the post-transcriptional circuitry via CNOT3 as a therapeutic vulnerability in AML.

Dysregulation of normal gene expression programs drives cancer development[1]. Besides transcriptional and epigenetic factors, post-transcriptional and translational control has been recognized as key regulatory mechanisms essential for malignant transformation and progression[2]. Somatic mutations and aberrant expression of proteins involved in multiple post-transcriptional processes have been identified in cancers, in particular in hematological malignancies[3–5] and mutations in the translational machinery have been found in genetic syndromes associated with increased susceptibility[6]. A number of key oncogenic pathways, such as c-MYC[7], RAS[8], and PIK3CA[9] have been found to alter the translational machinery to direct the production of proteins favouring survival and expansion of tumors. Abnormal pools

[1]Faculty of Pharmaceutical Sciences, University of British Columbia, Vancouver, Canada. [2]Terry Fox Laboratory, British Columbia Cancer Research Centre Vancouver, Vancouver, Canada. [3]Department of Experimental Medicine, University of British Columbia, Vancouver, Canada. [4]CRUK Beatson Institute, Glasgow, UK. [5]School of Cancer Sciences, University of Glasgow, Glasgow, UK. [6]Department of Medicine, Indiana University Simon Comprehensive Cancer Center, Indianapolis, IN, USA. [7]Genome Sciences Centre, British Columbia Cancer Research Centre, Vancouver, Canada. [8]Bioinformatics program, University of British Columbia, Vancouver, Canada. [9]Memorial Sloan Kettering Cancer Center, New York, NY, USA. [10]Department of Medical Genetics, University of British Columbia, Vancouver, Canada. [11]Department of Blood and Marrow Transplant and Cellular Immunotherapy, Moffit Cancer Center, Tampa, FL, USA. [12]These authors contributed equally: Maryam Ghashghaei, Yilin Liu. ✉e-mail: ly.vu@ubc.ca

of tRNAs have been found to facilitate optimal translation to promote gene expression programs wired by cancer cells to evade physiological restrictions and support malignant growth and disease progression[10–12]. However, while the general importance of translational regulation in cancer is established, the precise impact of an altered translation program on various aspects of tumorigenesis remains unclear, and the principal regulators of these processes have not been thoroughly elucidated.

Translation selectivity and efficiency are influenced by many parameters. These include mRNAs' abundance, sequences and modifications, availability of components involved in translation processes i.e., pools of ribosomes and specialized ribosomes, availability of amino acids and tRNAs, translation initiation, elongation and termination factors, and RNA regulating proteins as well as modifications of rRNAs and tRNAs[13–18]. The degree by which each of these factors dictates the protein synthesis rate is highly dependent on biological settings. Distinct translation programs have been found to drive cellular functions favoring proliferation vs. differentiation[10]. Optimizing codon usages in response to the availability of tRNAs and amino acids is central to maintaining translation efficiency[19]. In fact, the usage and frequency of synonymous codons, which are characterized by the presence of G/C vs. A/U at the wobble base position of a codon and its corresponding tRNAs, strongly affect translation efficiency to promote growth during cancer development[20,21]. In addition, translation is closely linked with other post-transcriptional processes, in particular mRNA decay to control and coordinate gene expression[22]. Hence, it is critically important to define the most relevant regulators specific to a given context and link the functional networks to the corresponding gene expression programs and phenotypic changes.

The multi-subunit CCR4-NOT complex has emerged as a key player in post-transcriptional control, participating in multiple regulatory steps of gene expression[23–26]. The highly conserved CCR4-NOT complex consists of several accessory proteins and two main modules i.e., the NOT scaffolding module comprising of CNOT1 and modular proteins CNOT2, CNOT3 and the catalytic module consisting of two distinct deadenylase classes (CNOT6/6 L and CNOT7/8)[27,28]. The CCR4-NOT complex is one of two major multi-subunit poly(A) deadenylation complexes, which mediates the shortening of the poly(A) tails and starts the mRNA decay process[29–32]. The CCR4-NOT complex can be recruited to specific target mRNAs via its interaction with the RNA-inducing silencing complex (RISC)[33] and RNA binding proteins (RBPs)[34,35]. Recent studies revealed that the CCR4-NOT complex can directly modulate translation elongation by sensing the decoding activity of translating ribosomes and coupling RNA degradation with ribosome processivity at non-optimal codons[36,37]. In yeast, it is demonstrated that the direct association with ribosomes is mediated by the subunits Not5/CNOT3 and CNOT4[36]. Despite being at the center of these critical processes, functions of the CCR4-NOT complex in cancer are poorly understood. In the present study, we uncovered CNOT3, a non-catalytic subunit of the CCR4-NOT complex, to be essential for leukemogenesis. Acute myeloid leukemia (AML) is a genetically complex diseases characterized by abnormal expansion and differentiation blockade of myelopoiesis resulting in an excessive accumulation of blasts and a suppression of normal blood cell production[38]. Multiple post-transcriptional mechanisms have been shown to contribute to pathogenesis of AML[5,39]. Here, we defined CNOT3 function in translation control to drive expression of a growth-promoting gene expression program, thus stimulating leukemogenesis.

## Results

### Expression of the subunit CNOT3 of the CCR4-NOT complex is elevated in AML

To identify novel regulators of post-transcriptional and translation control in AML, we surveyed several genome-wide CRISPR screens and observed that several subunits in the RNA deadenylation CCR4-NOT complex i.e. CNOT1, CNOT2, and CNOT3, and CNOT10 were highly ranked among genes essential for survival of human[40] and mouse[41] leukemia cells (Fig. 1A and Supplementary Fig. 1A). The enzymatic subunits of the complex (CNOT6/6 L/7) showed no apparent requirement, suggesting that they may play a redundant role or the requirement of other subunits in the complex is independent of its deadenylase activity. Interestingly, the poly(A) Nuclease-PAN2/PAN3 deadenylation complex was ranked more modestly on the screen, suggesting a more dominant role of the CCR4-NOT complex in AML. Among those subunits, CNOT3, the highest ranked among all components (#349 among 18,663 genes tested and consistently in all 14 AML cell lines tested in the study[40]), was found to be highly expressed across a panel of AML cell lines (Supplementary Fig. 1B). Furthermore, we observed that CNOT3 expression is elevated in the majority of assayed AML patient samples (10 out of 12) by immunoblot compared to normal human cord blood CD34+ cells (CB-CD34+ cells) (Fig. 1B and Supplementary Fig. 1C and Supplemental dataset 1). To evaluate CNOT3 expression in additional primary samples, we optimized the intracellular flow cytometry assay for the detection of endogenous CNOT3 protein (Fig. 1C and Supplementary Fig. 1D and Supplementary Dataset 1). We demonstrated that 13 out of 16 additional AML samples exhibited elevated CNOT3 expression in comparison with bone marrow samples obtained from healthy donors (Fig. 1D). These data suggested a potential role of dysregulated CNOT3 in AML.

### CNOT3 promotes survival and growth of leukemia cells

To directly examine the function of CNOT3 in AML, we depleted CNOT3 using two independent shRNAs (KD-33 and KD-37). With efficient knockdown of CNOT3 (Fig. 1E), we observed significant inhibition of growth across a panel of genetically diverse leukemia cell lines including MOLM13 (AML French-American-British classification (FAB) M5a; MLL-AF9 fusion, FLT3-ITD); OCI-AML3 (FAB M4; NPM1 and DNMT3 mutations); NB4 (APL/FAB M3; PML-RARA fusion); HL-60 (FAB M2; c-MYC amplification); NOMO-1 (FAB M5a; MLL-AF9 fusion) and THP-1 (pediatric AML; MLL-AF9 fusion) (Fig. 1F and Supplementary Fig. 1E–I). Knockdown of CNOT3 induced differentiation in leukemia cells evidenced by elevated expression of CD11b and CD14 cell surface markers (Fig. 1G) and changes in cellular morphology (Fig. 1H) and increased apoptosis (Fig. 1I). To complement the shRNA-mediated knockdown approach, we also performed sgRNA-mediated knockout of CNOT3 in two cell lines constitutively expressing Cas9 i.e., MV4-11 Cas9 and MOLM13 Cas9. Effective depletion of CNOT3 using sgRNAs (Supplementary Fig. 1J) also resulted in reduced cell proliferation (Supplementary Fig. 1K and Supplementary Fig. 1O); increased myeloid differentiation (Supplementary Fig. 1L, M and Supplementary Fig. 1P) and marked induction of apoptosis (Supplementary Fig. 1N and Q). On the other hand, we overexpressed CNOT3 in MOLM13, NB4, and THP-1 cells and observed that increased CNOT3 expression significantly augmented the growth of leukemia cells (Fig. 1J, K and Supplementary Fig. 1R, S). We also were able to rescue the growth defect in CNOT3-depleted cells by forced expression of CNOT3 (Supplementary Fig. 1T), indicating that the observed phenotypes upon CNOT3 depletion were on target. Taken together, these results strongly indicated that high CNOT3 expression promotes the survival of AML cells.

### CNOT3 is essential for leukemogenesis

To test whether CNOT3 is required for leukemogenesis, we injected leukemia cells transduced with control (scramble shRNA) and shRNAs (KD-33 and KD-37) against CNOT3 into NSG immunodeficient mice and followed leukemia development in vivo. We observed a marked reduction in the ability of leukemia cells to engraft upon CNOT3 depletion (Fig. 1L). Loss of CNOT3 significantly delayed leukemia development and improved survival of recipient animals (Fig. 1M and Supplementary Fig. 1U).

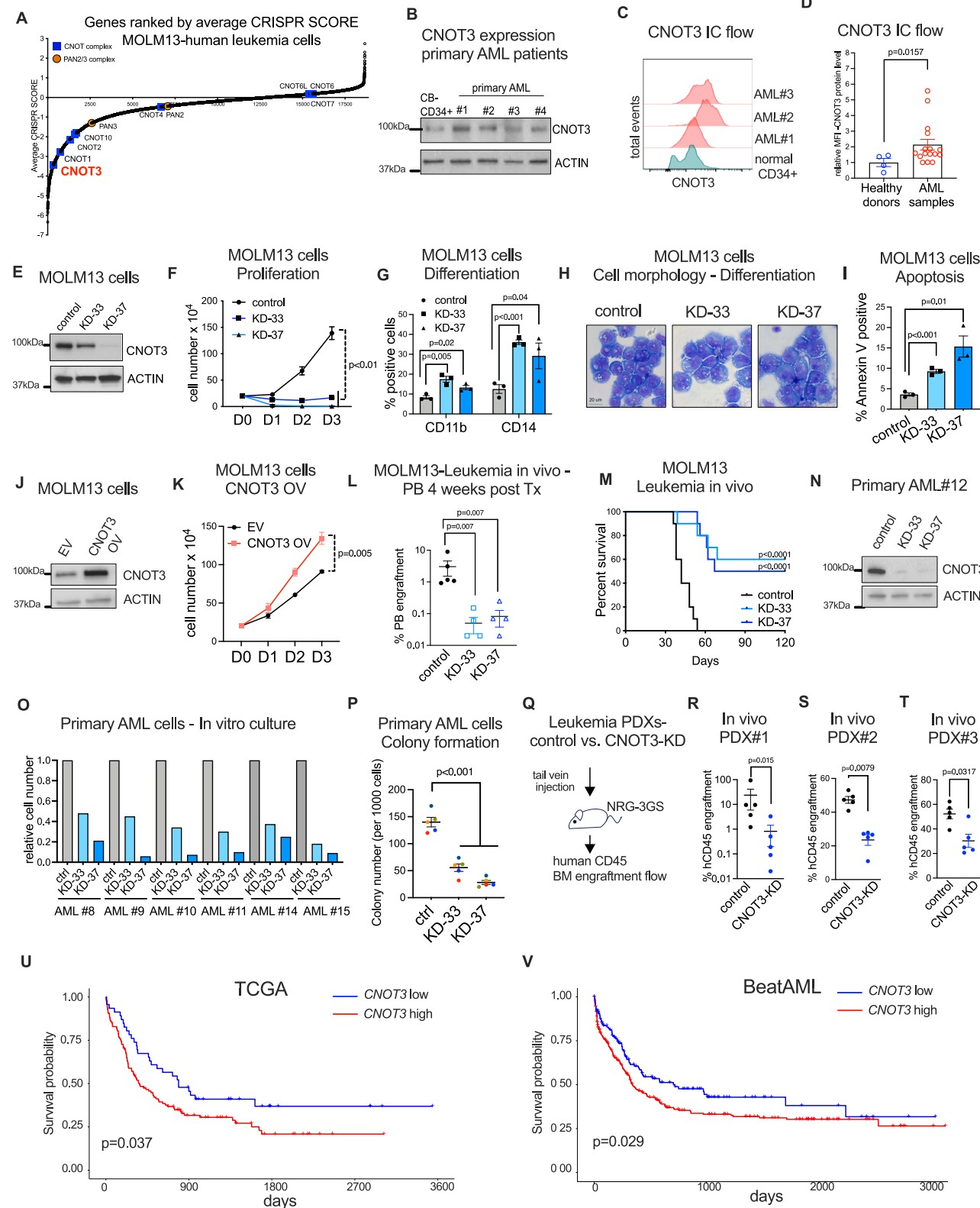

Next, to establish the relevance of CNOT3 in primary diseases, we depleted CNOT3 in primary AML cells using concentrated viruses expressing the control vs. KD-33 and KD-37 shRNAs and obtained efficient knockdown (Fig. 1N). In a panel of AML primary patient samples (Supplemental dataset 1), we found that CNOT3 loss of function decreased cell survival in liquid culture and markedly reduced their colony-forming ability (Fig. 1O, P and Supplementary Fig. 1V). In addition, using three patient-derived AML xenotransplantation (PDX)

models (Fig. 1Q and Supplemental dataset 1), we demonstrated that depletion of CNOT3 significantly hindered engraftment and disease progression of leukemia cells in vivo (Fig. 1R–T and Supplementary Fig. 1W). Importantly, we observed comparable CNOT3 expression levels in control vs. CNOT3-shRNA-transduced leukemic cells collected from mice succumbed from leukemia (Supplementary Fig. 1X); indicating that CNOT3 depleted cells were negatively selected against in vivo and leukemia outgrew from the CNOT3-knocked down samples

**Fig. 1 | CNOT3 is essential for leukemogenesis. A** CRISPR score rank for essentiality in MOLM13 leukemia cells of subunits of deadenylation complexes−CCR4-NOT complex and PAN2/PAN3 complex. **B** Immunoblots showing CNOT3 protein in CB-CD34+ cells and primary AML patient cells. **C** Representative flow plots showing detection of endogenous CNOT3 in healthy donors vs. primary AML patient cells (AML samples). **D** Quantification of median fluorescent intensity (MFI) of CNOT3 levels in healthy donors ($n = 4$) vs. primary AML patient cells ($n = 16$). Data shown as mean ± s.e.m, $p$ value calculated by Mann−Whitney test. **E–I** MOLM-13 expressing either a scramble (control) shRNA or CNOT3-targeting shRNAs (KD33 and KD-37). **E** Immunoblots showing efficient knock down of CNOT3. **F** Cell proliferation. **G** Quantitative summary of flow cytometry analysis of myeloid markers CD11b and CD14. **H** Representative H&E images. **I** Percentage of Annexin-V positive cells by flow cytometry. **J**, **K** MOLM-13 cells expressing empty vector (control) or cDNA expressing CNOT3 (CNOT3-OV). **J** Representative immunoblots confirming CNOT3 overexpression. **K** Cell proliferation. Graphs (**F**, **G**, **I**, **K**) showing data as mean ± s.e.m, $n = 3$ independent experiments, $p$ values calculated by two-tailed student's $t$ test. **L** Percentage of human CD45 positive cells in peripheral blood (PB) of NSG recipient mice 4 weeks post transplantation. Control $n = 5$; KD-33 $n = 4$; KD-37 $n = 4$. mean ± s.e.m, $p < 0.001$, Mann−Whitney test. **M** Kaplan−Meier curves. Control $n = 10$; KD-33 $n = 10$; KD-37 $n = 10$. $p < 0.001$, Log-rank test. **N** Immunoblots showing knockdown of CNOT3 $n = 1$ sample. **O** In vitro culture $n = 6$ primary samples and **P** Colony formation. $n = 5$ primary samples. Data shown mean ± s.e.m., $p < 0.001$, two-tailed Student's $t$ test. **Q** Experimental scheme to assess leukemia in vivo. **R–T** Percentage of human leukemia CD45+ cells in recipient animals in three independent PDX models. $N = 5$ per group. mean ± s.e.m, $p$ values calculated by Mann−Whitney test. **U**, **V** High expression of *CNOT3* mRNA correlates with poor prognosis in AML patients. Kaplan−Meier curves showing outcomes of AML patients in **U** TCGA-AML high $n = 117$ vs. low $n = 46$; **V** Beat AML high $n = 271$ vs. low $n = 141$. $p$ values calculated using Log-rank test. Source data are provided as Source Data files for figures (**B**, **D–G**, **I–T**).

were driven by cells successfully evaded shRNA-mediate gene silencing. To further evaluate the importance of CNOT3 function in human AML, we surveyed CNOT3 expression and found that high CNOT3 expression correlates with poor prognosis of AML patients across three independent AML patient cohorts i.e. German AMLCG 1999 trial[42] (Supplementary Fig. 1Y); The Cancer Genome Atlas Program[43] (TCGA) (Fig. 1U) and beat AML[44] (Fig. 1V). Taken together, the data established a critical role for CNOT3 in leukemia development.

### CNOT3 maintains undifferentiated states of human hematopoietic stem/progenitor cells

We next want to examine whether CNOT3 is differentially required between leukemia cells vs. normal hematopoietic stem/progenitor cells (HSPCs). Based on the public dataset of human hematopoietic cells (Bloodspot[45]), *CNOT3* is highly expressed in stem cell compartment and downregulated in progenitors and differentiated cells in human hematopoietic hierarchy (Supplementary Fig. 2A). Interestingly, CNOT3 expression is further downregulated from myeloid progenitors to myeloid cells but not from lymphoid progenitors to B or T cells, suggesting CNOT3 might function in a lineage-specific manner. To delineate CNOT3 role in normal HSPCs, we knocked down CNOT3 in cord blood-derived CD34+ HSPCs using the KD-33 and KD-37 hairpins (Fig. 2A). While we observed a noticeable reduction in cell numbers in the KD-37 condition, the effects on cell proliferation using both shRNAs in HSPCs were considerably milder in comparison to the complete abrogation of cell growth observed with leukemia cells (Fig. 2B compared with Fig. 1F and Supplementary Fig. 1E–I). Loss of CNOT3 resulted in reduced colony formation (Fig. 2C). However, CNOT3 depletion did not induce apoptosis in HSPCs, pointing to the differential response when compared to leukemia cells (Fig. 2D and Supplementary Fig. 2B). On the other hand, we found strong induction of myeloid differentiation upon CNOT3 depletion. HSPCs gained marked increased expression of several myeloid markers CD11b, CD14 and CD13 (Fig. 2E and Supplementary Fig. 2C) and exhibited clear morphological changes corresponding to mature myeloid cells (Fig. 2F). Conversely, overexpression of CNOT3 enhanced expansion of HSPCs in liquid culture (Fig. 2G), increased colony forming ability (Fig. 2H) and significantly blocked myeloid differentiation of HSPCs (Fig. 2I and Supplementary Fig. 2D). The data strongly indicated that CNOT3 functions to maintain HSPCs in an undifferentiated stage.

### Saturated domain screening revealed the NOT box domain to be critical for CNOT3 function in leukemia cells

CNOT3 has been shown to be involved in several layers of gene expression control including transcription, modulation of CCR4-NOT deadenylase activity, and direct sensing of codon optimality, which is shown to be mediated by distinct functional domains[35]. The major isoform of human CNOT3 protein (UniProtKB-O75175) consists of 753 amino acids with an N-terminal coiled-coil domain (CNOT3-N-term); a

linker region (CNOT3-Middle) and the C-terminal regions consisting of a NOT1 anchor region (NAR), a connector sequence (CS) and a NOT-box domain[27]. Thus, to determine the most functionally relevant domains of CNOT3 in AML, we employed a CRISPR/Cas9 saturation mutagenesis screen by tiling the entire CNOT3 gene with sgRNAs (Fig. 3A). Small deletions induced by Cas9 can create in-frame mutations in targeted regions without affecting expression of the full-length protein[46,47]. A pool of viruses expressing control non-targeting and safe-harbor sgRNAs and the library of 464 sgRNAs targeting all available Cas9 recognition sequences in CNOT3 was transduced into MOLM13 constitutively expressing Cas9 (Supplemental dataset 2). DNA was collected 48 hours after transduction (D0) and after 10 doubling times (D14) for sequencing to identify sgRNAs enrichment (Supplemental dataset 3 and Supplementary Fig. 3A–D). CRISPR scores were calculated as the ratio of sgRNA abundancy at D0 vs. D14. We obtained specific enrichments of CNOT3-targeted sgRNAs vs. control sgRNAs (non-targeting sgRNAs or sgRNAs targeted to genomic safe-harbor regions) (Supplementary Fig. 3E). We then used the CRISPRO program[48] to smooth the CRIPSR score via a LOESS (LOcally WEighted Scatter-plot Smoother) regression to model the local trends of the perturbation effect and generate the annotated graph for the entire protein. The analysis yielded two regions: the majority of the N-term domain and most significantly the entire NOT box domain in the C-term (Fig. 3B). It is recently reported that CNOT3 can directly interact with stalled ribosomes via its N-term domain while binding to several subunits of the CCR4-NOT complex including CNOT1, CNOT2, CNOT4, CNOT6 and CNOT10/11 via both the N-term domain and the C-term domain[49]. The interaction is critical for the CCR4-NOT complex's ability to recognize slow-moving ribosomes. Therefore, the results suggested a functional dependence of AML cell viability on codon sensing and translational control via CNOT3 and the CCR4-CNOT complex.

To validate the CRISPR screen finding, given that mutations introduced at the C-terminal area are more likely to be specific for the domain, we focused on the NOT box domain and created the NOT box-truncated mutant of CNOT3 (NOT box truncated). We overexpressed the mutant in MOLM13 cells in parallel with the full-length CNOT3 (CNOT3 FL) (Fig. 3C). We found that deletion of the NOT box domain abolished the growth advantage observed in the CNOT3-FL overexpressing cells (Fig. 3D). To dissect the downstream gene expression programs modulated by CNOT3 and in particular the NOT box domain, we profiled transcriptomic changes in CNOT3-FL and NOT box truncated cells by RNA-sequencing. Interestingly, RNA-sequencing analysis revealed a complete abrogation of transcriptomic changes induced by CNOT3 overexpression when the NOT box domain was removed (Fig. 3E vs. F and Supplemental dataset 4, 5). Overexpression of CNOT3 resulted in a shift in suppression of gene expression (Fig. 3E), and geneset enrichment analysis (GSEA)[50] demonstrated significant upregulation of the oxidative phosphorylation pathway and targets of

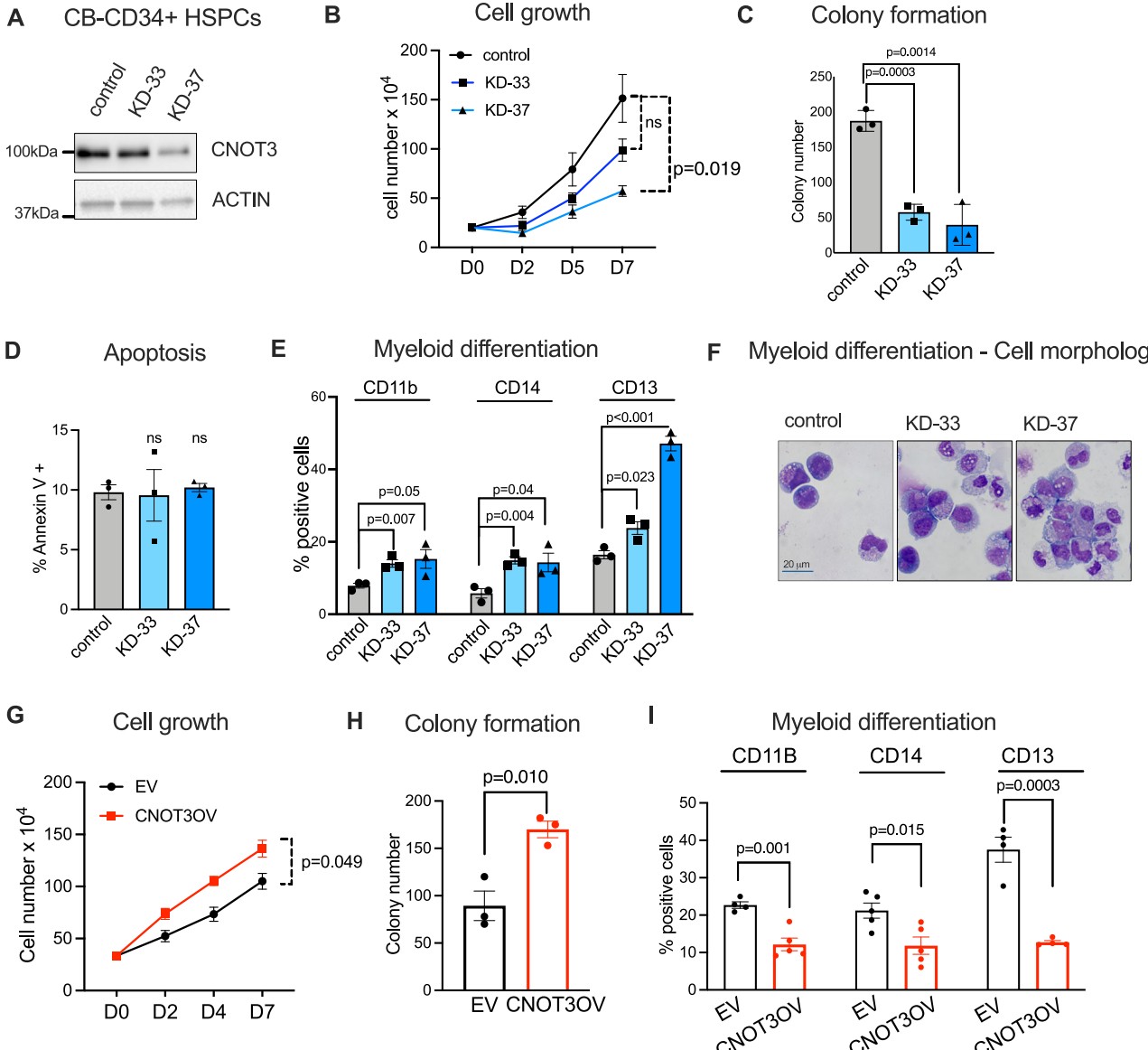

**Fig. 2 | CNOT3 suppress differentiation of primary HSPCs. A–F** Cord blood-derived human CD34+ hematopoietic stem/progenitor cells (CB-CD34+ cells) were transduced with lentiviruses expressing either a scramble (control) shRNA or *CNOT3*-targeting shRNAs (KD33 and KD-37). Cells were selected for puromycin resistance and assayed 3 days post-transduction. **A** Representative immunoblots showing efficient depletion of CNOT3. ACTIN serves as loading control. **B** Cell proliferation in liquid culture supplemented with cytokines. **C** Colony formation. Cells were plated on methylcellulose supplemented with cytokines. **D** Percentage of apoptotic cells by flow cytometry analysis for Annexin-V positivity. **E** Quantitative summary of flow cytometry analysis of myeloid differentiation markers CD11b, CD14, and CD13. **F** Representative H&E images of morphological evaluation of CB-CD34+ cells upon CNOT3 depletion. All graphs (**A–E**) show data as mean ± s.e.m. *n* = 3 independent experiments, *p* values calculated by two-tailed student's *t* test. **G–I** CB-CD34+ cells were transduced with lentiviruses carrying either an empty vector backbone or cDNA expressing *CNOT3*. Reporter gene YFP is present within the lentiviral vector Cells were sorted based on YFP positivity 3 days post-transduction. **G** Cell proliferation in liquid culture supplemented with cytokines. **H** Colony formation. Cells were plated on methylcellulose supplemented with cytokines. **I** Quantitative summary of flow cytometry analysis of myeloid differentiation markers CD11b, CD14, and CD13. All graphs **G–I** showing data as mean ± s.e.m, *n* = 3 independent experiments, *p* values calculated by two-tailed student's *t* test. Source data are provided as Source Data files for figures (**A–E, G–I**).

c-MYC (Fig. 3G, H) and downregulation of the KRAS signaling and coagulation pathways (Fig. 3I, J). Overall, these data indicated that the NOT box domain is required for CNOT3 function in promoting the growth of leukemia cells.

## CNOT3 regulates c-MYC expression at the translational level

To uncover the molecular mechanisms underlying CNOT3 function in AML, we sought to characterize the changes in gene expression programs upon CNOT3 loss of function. Given its role in post-transcriptional gene expression control, we performed both transcriptomic and proteomic profiling of control vs. shRNA mediated *CNOT3* knockdown leukemia cells at ~2 days post transduction, the

earliest time point when we obtained a consistent reduction of CNOT3 expression using the two KD-33 and KD-37 shRNAs (Fig. 4A). We observed robust changes in CNOT3 depleted cells at both the transcriptomic and proteomic levels (Supplementary Fig. 4A, B and Supplemental dataset 6–9). GSEA analysis of both omics analyses showed a preferential reduction in the expression of genes involved in G2/M checkpoint and targets of c-MYC and increased expression of genes enriched in IFN-γ response and p53 pathways (Fig. 4B, C). CNOT3 had been shown to influence p53 function by controlling p53 mRNA stability[51]. While we did not observe changes in p53 itself, both RNA-seq and mass spectrometry analyses captured upregulation of CDKN1A (highlighted in Supplementary Fig. 4A, B), one of

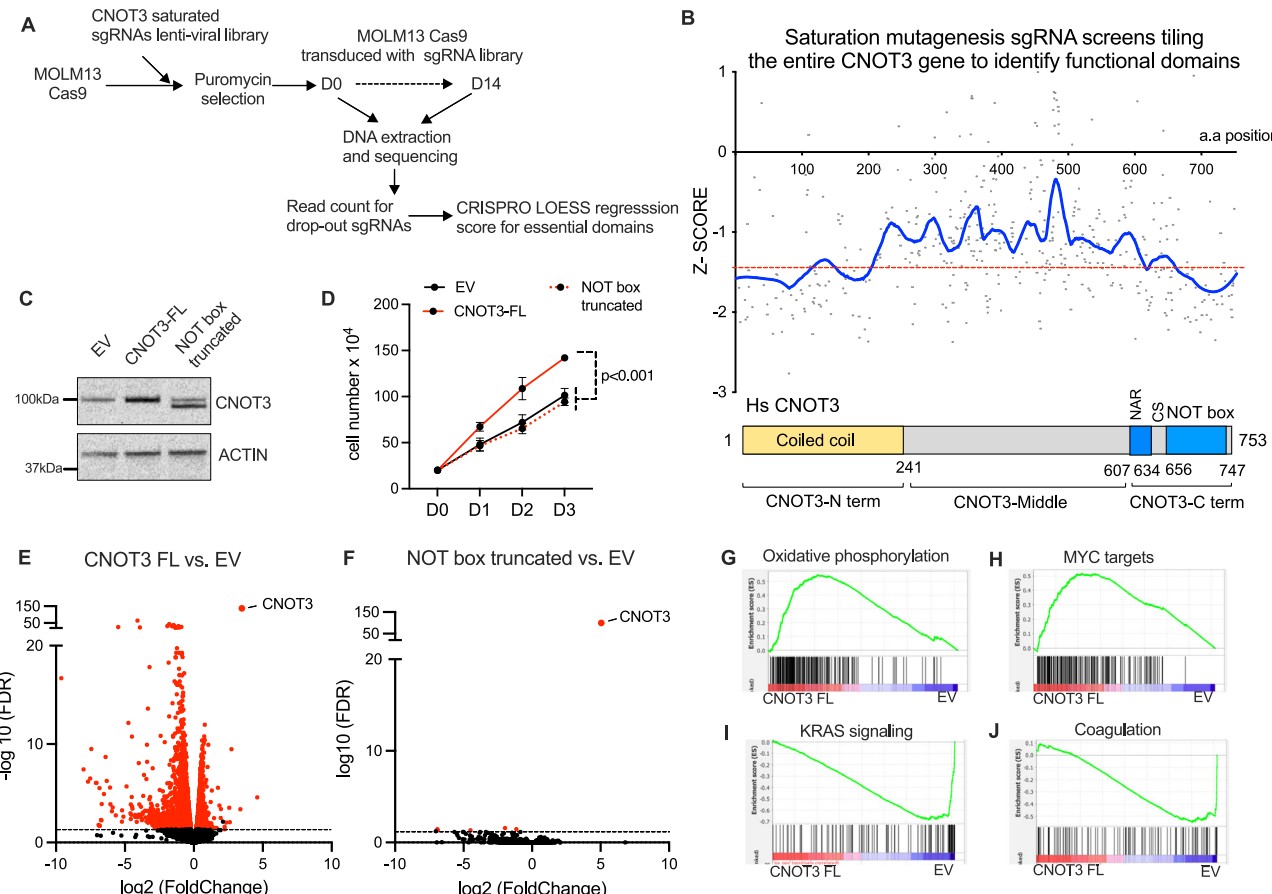

**Fig. 3 | NOT box domain is essential for CNOT3 function in AML. A** Experimental scheme of CRISPR/Cas9 tiling sgRNA mutagenesis screen for essential domains of CNOT3 in MOLM13 myeloid leukemia cells. **B** Top- Graphs presenting z score of individual sgRNAs and smoothed out the distribution of score for functional domains. Read counts for drop-out and/or enrichment of each sgRNA were used to calculate functional scores of gRNAs to specific protein domains using the CRISPRO pipeline. LOESS regression was used to smooth the tiling signals from individual sgRNAs (gray dots) to identify essential amino acids and domains. Bottom–Annotations of CNOT3 structural domains. **C** Immunoblots showing expression of full-length CNOT3 (CNOT3-FL) and NOT box-truncated proteins. ACTIN serves as loading control. **D** Cell proliferation of MOLM13 cells transduced with empty vector (EV) control, cDNA expressing full-length *CNOT3* or NOT box-truncated *CNOT3*. Data shown as mean ± s.e.m., n = 3 independent experiments, p < 0.001, two-tailed Student's t test. **E, F** Volcano plots showing the distribution of differentially expressed genes upon overexpression of **E** Full-length CNOT3 (CNOT3-FL) and (**F**) NOT box-truncated CNOT3 in comparison to empty vector control. **G–J** Gene set enrichment analysis (GSEA) of RNA-seq dataset shown in **E**. Source data are provided as Source Data files for figures (**B**–**D**).

the direct targets of p53, linking our results with the previous report[51].

Given the important role of c-MYC in cancer and leukemogenesis, we further investigated the functional relationship between CNOT3 and c-MYC. We systematically evaluated the consequences of CNOT3 loss of function on *c-MYC* mRNA and c-MYC protein expression at different regulatory steps of gene expression control. We observed that CNOT3 depletion (by both shRNAs and sgRNAs) resulted in marked reduction of c-MYC protein in many cell lines (Fig. 4D and Supplementary Fig. 4C–F). We found no significant change in total *c-MYC* mRNA abundance or level of primary *c-MYC* transcript, thus ruling out the control at the transcriptional step (Fig. 4E). Because the CCR4-NOT complex is known to mediate mRNA degradation, we examined *c-MYC* mRNA decay upon CNOT3 depletion by tracing mRNA levels after treating cells with actinomycin D to block transcription. We observed no substantial reduction in *c-MYC* half-life (Fig. 4F). We also checked for potential impacts on RNA export between nucleus vs. cytoplasm (Supplementary Fig. 4G–K). We confirmed by immunofluorescence (IF) analysis reduction of c-MYC protein (Supplementary Fig. 4H, I) and no change in *c-MYC* mRNA level (Supplementary Fig. 4J) and showed that CNOT3 depletion also did not alter cellular distribution of *c-MYC* transcripts (Supplementary Fig. 4K). In addition, no change in protein half-life or protein degradation rate

was observed upon CNOT3 knockdown (Supplementary Fig. 4L, M). Finally, we evaluated the impact of CNOT3 loss on translation by performing polysome profiling of control vs. CNOT3 depleted cells (KD-33 and KD-37). We found similar profiles across all three samples, indicating that there was no gross defect in global translation (Fig. 4G). Interestingly, when we examined abundance of *c-MYC* transcripts from the monosome vs. polysome enriched fractions which correspond to less active vs. more active translation activities, we found a significant decrease in abundance of *c-MYC* mRNAs associated with polysomes (Fig. 4H). The results suggested that CNOT3 might facilitate the recruitment of *c-MYC* transcripts to active polysomes to drive high expression levels of c-MYC protein in AML.

## Global analysis revealed a dependent correlation of the GC content and specific codons with transcripts' abundance

Next, we investigated whether there are mRNA features that preferentially specify their dependency on CNOT3 for gene expression. We defined the three groups of genes whose levels are downregulated (FDR ≤ 0.05, log2FC < 0); up-regulated (FDR ≤ 0.05, log2FC > 0); and unchanged (FDR > 0.05) in our RNA-seq experiment (described in Fig. 4 and Supplementary Fig. 5A). In order to ascertain feature properties of transcripts regulated by CNOT3, we undertook a supervised machine learning approach in the form of gradient boosting (train

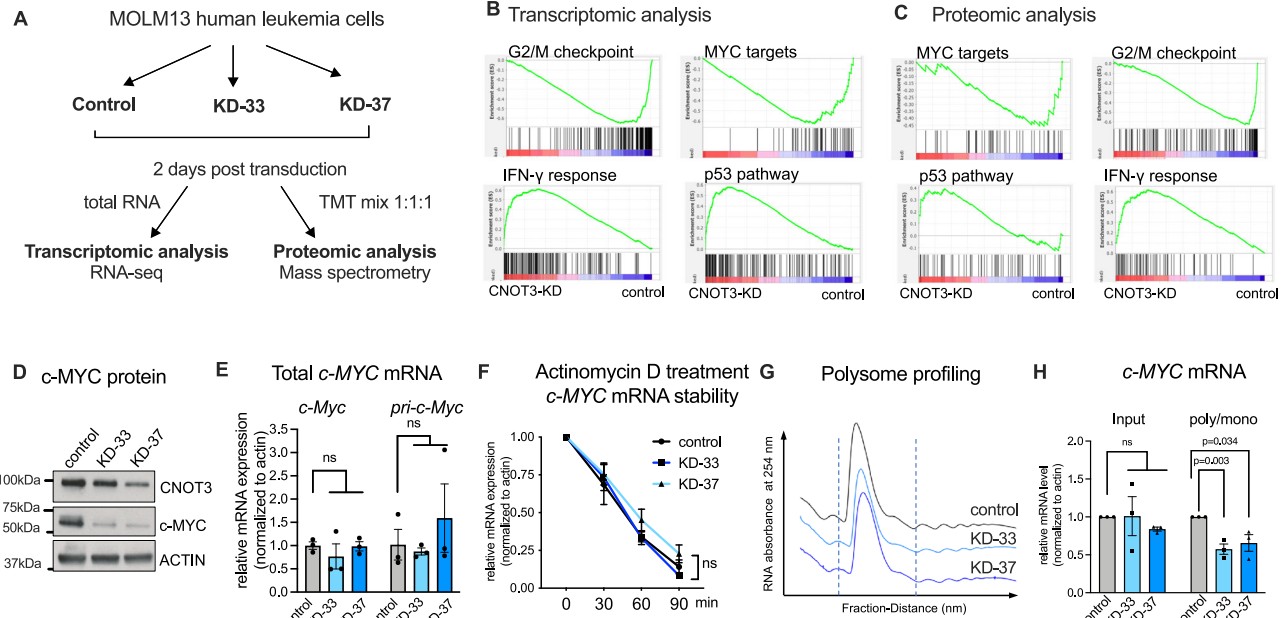

**Fig. 4 | CNOT3 controls translation of c-MYC. A** Experimental scheme of omic profiling by RNA-sequencing and proteomic analysis by tandem mass tag (TMT) mass spectrometry. MOLM-13 cells were transduced with lentiviruses expressing either a scramble (control) shRNA or *CNOT3*-targeting shRNAs (KD33 and KD-37). Cells were selected for puromycin resistance and assayed 2 days post-transduction. *n* = 3 independent experiments. **B, C** Gene set enrichment analysis (GSEA) of **B** transcriptomic profiling and **C** proteomic analysis of CNOT3 depleted cells vs. control as described in **A. D** Immunoblot confirming efficient knockdown of CNOT3 and reduction of c-MYC upon CNOT3 depletion in MOLM13 cells. ACTIN serves as loading control. **E** Quantitative qRT-PCR assessment of mature *c-MYC* transcript and primary (unprocessed) *c-MYC* transcript in MOLM13 cells.

**F** Quantitative qRT-PCR measure of *c-MYC* abundance over time after treatment of cells with actinomycin D to inhibit transcription. *ACTIN* serves as housing keeping gene control. **G** Polysome profiling of CNOT3 depleted cells vs. control. Graphs showing optical density profiles (ODA$_{254}$) of RNA across polysome gradients. Mono di-some and polysome fractions are shown. **H** Quantitative qRT-PCR measure of *c-MYC* abundance in input and relative abundance in polysome vs. monosome fractions. All *n* = 3 independent experiments, two-tailed Student's *t* test, ns not significant. All graphs show data as mean ± s.e.m, *n* = 3 independent experiments, *p* values calculated by two-tailed student's *t* test. Source data are provided as Source Data files for figures (**D–F, H**).

AUC: 0.72, test AUC: 0.68; Supplementary Fig. 5B, C) for a number of attributes that may influence gene expression including length, sequence and nucleotide composition[37]. Top contributors to this model included the percentage of A and G nucleotides in the CDS (CDS AG%), percentage of G and C nucleotides in the 5′UTR (5′UTR GC%), the length of the CDS (CDS length), the length of the 3′UTR (3′UTR length) and the percentage of codons with a G or a C at the wobble-3$^{rd}$ base position (GC3%) (Fig. 5A and Supplementary Fig. 5D–J). Whilst statistically significant trends were observed for coding sequence (CDS) AG % (Supplementary Fig. 5D), CDS length (Supplementary Fig. 5F), and 3′ UTR length (Supplementary Fig. 5G) and 5′UTR GC% (Supplementary Fig. 5H), the differences were small, suggesting an overall contribution of multiple factors.

However, given that CNOT3 has recently been shown to involve in monitoring codons[36], we decided to further examine the GC vs. AU content at the 3rd position of the codon (GC3 vs. AU3). Indeed, downregulated transcripts (upon CNOT3 knockdown) appeared to be less GC3-rich when compared with upregulated transcripts (Fig. 5B, C). It has been previously described that distinct codon usages in proliferation versus differentiation-associated genes are characterized by an AU3 and a GC3 bias, respectively[10,20]. Interestingly, relative synonymous codon usage analysis demonstrated that the most abundant transcripts in MOLM-13 leukemia cells exhibit an AU3 bias, suggesting an AU3 polarization of the cell's transcriptome corresponding with the proliferative state of the cells (Fig. 5D). However, transcripts upregulated upon CNOT3 depletion are characterized by a GC3 bias, which may be due to dysfunctional recruitment of the CCR4-NOT complex to ribosome stalling at non-optimal codons (Fig. 5E). These data are consistent with previous observation of selectively stabilized G/C ending codons upon knockdown of CNOT1, the central scaffold of the

CCR4-NOT complex[37]. Enrichr hallmark pathway enrichment[52] analyses further revealed that downregulated transcripts were enriched in pathways more biased towards an AU3 vs. GC3 codon signature, particularly transcripts in growth-promoting pathways including E2F targets, G2-M checkpoint, c-MYC targets (Fig. 5F). These results corroborated our phenotypic observations of the critical requirement of CNOT3 for leukemia growth and survival while inhibiting differentiation in both normal and leukemia cells.

To further look into the link between CDS sequence composition and CNOT3-mediated gene expression, we examined the codon frequency and distribution along transcripts. We also looked for positional differences in codon usage across the transcripts for both the downregulated and upregulated groups for each decile of the CDS by normalizing to the unchanged group. While there is no strong positional effect, hierarchical clustering revealed that the AU3 codons formed a distinct cluster from the GC3 codons consistently along a transcript (Fig. 5G). Except for 4 codons (i.e., ATG, AAG, TGT, and CAT), we observed a clear segregation of clusters in which A/T ending codons are highly enriched while the G/C ending codons are less abundant in downregulated genes (Fig. 5G A/T green vs. G/C orange). Since amino acid content can influence processivity of ribosome in decoding and translating mRNAs, we next examined the frequency of codon triplets encoded amino acids in the two differentially regulated groups. We found tyrosine is overrepresented in the upregulated group, and lysine is highly enriched in the downregulated genes (Supplementary Fig. 5K). The presence of strings of lysine codons, in particular the AAA codons, has been shown to inhibit translation and induce instability of a mRNA transcript[53]. Interestingly, the AAA codon is the synonymous codon showing the highest usage in genes downregulated when CNOT3 is depleted (Fig. 5G). Comparing the biased

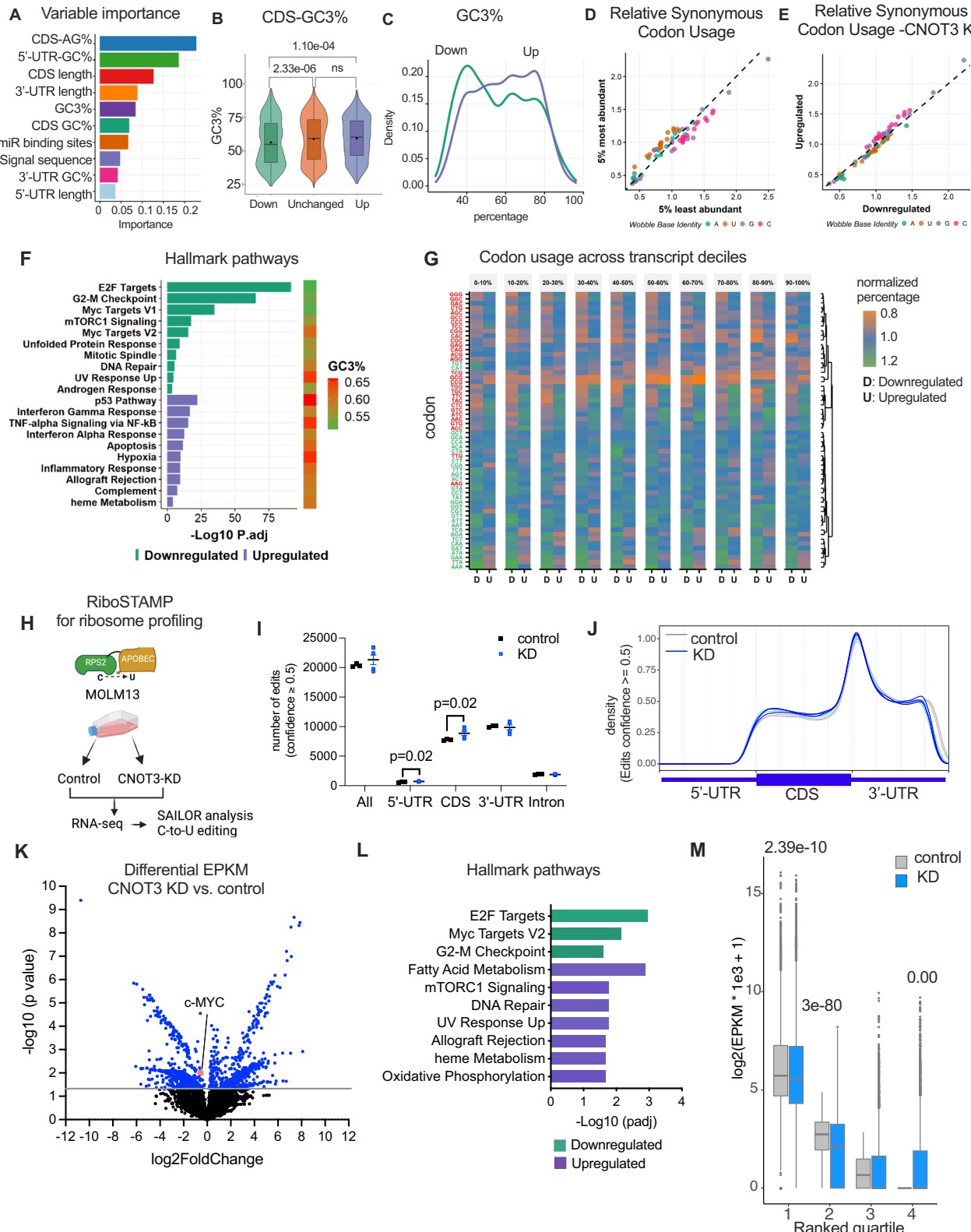

usage of codons and amino acids, we also observed differential usage of synonymous codons. For example, while leucine showed slightly different usage, the six synonymous codons exhibited opposite patterns i.e., AAT, GAT, and GTT are highly enriched with downregulated genes, while AAC, GAC, and GAG are not. Taken together, these results suggested that codon usage contributes to mediating CNOT3's control of gene expression.

## Loss of CNOT3 impacted translation efficiency

Given the potential involvement of CNOT3 in protein synthesis, we employed the RPS2– APOBEC1 (RiboSTAMP) method[54] (Fig. 5H) to examine the consequences of CNOT3 depletion on translational activity. We created a MOLM13 cell line carrying an inducible *RPS2-APOBEC1* prior to transducing the cells with control (scramble shRNA) and shRNAs against CNOT3 (KD-33 and KD-37). Cells were induced to

**Fig. 5 | Global assessments of CNOT3 mRNA targets. A** Summary of ranked importance score of variables determining transcripts upregulated ($n = 1760$); downregulated ($n = 1456$), or unchanged ($n = 15779$) upon CNOT3-KD. $n = 3$ control vs. CNOT3-KD (KD-33 and KD-37) independent biological samples. **B** Violin plot showing analysis of CDS-GC3 percentage in downregulated, unchanged, and upregulated genes. One-way ANOVA test, $p$ values denoted. The boxplots show the 0th (Q0) percentile, the 100th percentile (Q100), and black squares denote the mean. Q0 = 0.237, 0.213, and 0.255; Q100 = 0.965, 0.989, and 0.954 and the inter-quartile range = 0.294, 0.279, and 0.261 for downregulated, unchanged, and upregulated, respectively. **C** Density plot showing GC3 percentage. **D, E** Relative synonymous codon usage analysis of (**D**) top 5% most vs. least abundant genes in MOLM13 cells, **E** genes down- and upregulated upon CNOT3 knockdown. **F** Gene enrichment analysis showing enriched hallmark pathways and transcripts' GC3 contents. Fisher's exact test. **G** Heatmap showing enrichment of genetic codons presented in upregulated; downregulated, or unchanged genes. **H** Experimental

scheme of ribosome profiling by RiboSTAMP (RPS2-STAMP) method to evaluate translation activity in MOLM13 cells in control vs. CNOT3 KD. Image created using BioRender. **I** Total number of APOBEC-mediated editing events scored with ≥ 0.5 confidence level along the entire transcripts and within 5'UTR and CDS. Mann–Whitney tests, $p$ value denoted. Source data provided as Source Data Fig. 5I. **J** Metagenomic view of editing events across transcripts. **K** Volcano plot showing differential EPKM analysis CNOT3 KD vs. control. **L** Gene enrichment analysis of genes in **J**. Fisher's exact test. **M** Comparison of normalized EPKM values between control ($n = 3$) vs. CNOT3 depleted conditions ($n = 6$) in 4 quartile groups (graphs showing minima = 0; for each group control vs. KD: maxima, center, bounds of box (lower-25th percentile and higher- 75th percentile) and whiskers: group 1:68863 vs. 62139, 51.9 vs. 46.7, 24.9 and 153.1 vs. 18.7 and 147.6, 345.5 vs. 341; group 2: 28.5 vs. 293, 5.7 vs. 3.3, 2.8 and 9.3 vs. 0 vs. 8.5, 19 vs. 21.4; group 3: 6.1 vs. 979, 0.59 vs. 0, 0 and 1.8 vs. 0 and 2.09, 4.5 vs. 5.24; group 4: control all value = 0 and KD: 833, 0, 0 and 2.7, 6.8). Wilcoxon rank-sum one-sided test, $p$ values denoted.

express RPS2-APOBEC1 by doxycycline while underwent viral trans-duction and puromycin selection. Upon expression of RPS2-APOBEC1 fusion protein, ribosomal protein RPS2 directs the recruitment of RPS2-APOBEC1 with deaminase activity to the RPS2 binding sites. This results in a C-to-U base conversion (G-to-A in RNA-sequencing data), which serves as a readout for the presence of ribosomes along a mRNA. We obtained high-quality data with editing numbers consistent across replicates of all conditions (Supplementary Fig. 5L–N). Edited sites were scored, and only confident sites (confidence score ≥0.5) were included in downstream analysis. We observed an equivalent total number of edits in control vs. CNOT3 depleted cells, indicating that knockdown of CNOT3 did not affect global recruitment of ribosomes to mRNAs (Fig. 5I). Metagene analysis showed the previously described distribution of edit sites along mRNAs in both control vs. CNOT3 KD (Supplementary Fig. 5O)[54]. We noted there is a slight increase in edits in 5'UTR and CDS regions, suggesting potential retainment of ribosomes at these sites upon CNOT3 depletion (Fig. 5I).

To account for the influence of CNOT3 on mRNA abundance, we calculated the edited read counts per target as normalized values for length and coverage (EPKM) to incorporate ribosome-occupancy within CDS and mRNA levels to infer translation activity. To look at the regulation at the gene level, we performed differential EPKM analysis to identify genes whose translation activities are influenced by CNOT3 depletion (Fig. 5J and Supplemental dataset 10). In agreement with our polysome profiling data with c-MYC, we found c-MYC is among the translationally downregulated transcripts in CNOT3 KD. Interestingly, gene enrichment analysis revealed upregulation of translation of pathways, including fatty acid metabolism, mTORC signaling, DNA repair, and oxidative phosphorylation (Fig. 5K and Supplemental dataset 11). These pathways were not found to be enriched in tran-scriptomic changes upon CNOT3 depletion. On the other hand, loss of CNOT3 preferentially resulted in negative impact on translation activity of EF2 targets, c-MYC targets, and genes belonging to the G2-M checkpoint pathway (Fig. 5K and Supplemental dataset 12). The downregulated effects are in line with the alternations in tran-scriptomic profiles of CNOT3-depleted cells (Fig. 4F), suggesting that CNOT3 mediates translation activity to further enforce the expression of the growth-driven genes to promote proliferation and survival of leukemia cells.

Based on their EPKM values, each gene can be classified into dif-ferent groups with differential protein synthesis activity–higher EPKM correlates with more ribosome occupancy, hence higher translation efficacy. Therefore, we divided all profiled transcripts to 4 quartiles based on their EPKM in the unperturbed condition and asked whether CNOT3 loss of function affects the translation of these 4 groups dif-ferently. Interestingly, we observed that while there is no global change in translation (Figs. 3G and 5I), CNOT3 depletion significantly reduced EPKM of the top two quartiles, which represent the more actively translated mRNAs (Fig. 5L). The impact was not observed in

the lower quartiles, which in contrast exhibited more ribosome occu-pancy when CNOT3 was knocked down. The data supported the notion that CNOT3 might enhance translation efficiency of highly translated mRNAs to facilitate their protein production.

## CNOT3 associates with translation machinery in AML

A number of diverse molecular functions has been ascribed for CNOT3 as a subunit of the multi-functional CCR4-NOT complex. These include transcriptional regulation, RNA deadenylation, mRNA decay and monitoring codon optimality[28,55]. Thus, to determine which specific functions are important for CNOT3 role in AML, we performed immunoprecipitation of CNOT3 and mass spectrometry analysis of co-precipitated proteins to define CNOT3 interacting networks (Fig. 6A). 170 proteins were identified to be specifically associated with CNOT3 (uniquely pulled down with CNOT3 antibody or exhibited at least 2-fold enrichment in CNOT3 pulldown vs. IgG control) (Supplementary dataset 13). STRING network analysis showed that CNOT3 together with all members of the CCR4-NOT complex were recovered, con-firming the efficiency and specificity of the immunoprecipitation (Fig. 6B). In addition, we found abundant RNA associating protein and several proteins in the proteosome to be co-immunoprecipitated with CNOT3. Most significantly, there are two large clusters of ribosomal proteins and translation elongation factors (EEFs) (Fig. 6B). Notably, EEF1G, the enzymatic delivery of aminoacyl tRNAs to the ribosome and EEF1E, the auxiliary component of the multi-synthetase complex and several tRNA synthetases were among the proteins associating with CNOT3. Gene enrichment analysis by Enrichr (http://amp.pharm. mssm.edu)[56] also revealed that the majority of CNOT3 interacting proteins are involved in RNA metabolism and protein synthesis (Fig. 6C and Supplemental dataset 14). The interaction of CNOT3 with the translational machinery is well aligned with the results on CNOT3 structure-function (Fig. 3B) and the link between CNOT3 and AU3 vs. GC3 content of mRNAs (Fig. 5C–E), pointing to a dominant role in codon sensing and modulation of translation activity by CNOT3 in leukemia.

To further examine the association between CNOT3 and the elongation factors on actively translating mRNAs, we employed the Puromycin Labeling Coupled with Proximity Ligation Assay[57] (Puro-PLA), which can capture in situ interactions of macromolecules at distances less than 40 nm[30]. In the assay, leukemia cells were treated with puromycin to mark the elongating peptides, which serves as the surrogate indication of transcripts undergoing translation. Proximity ligation was then performed on secondary antibodies (anti-rabbit and anti-mouse) recognizing those raised against puromycin and our proteins of interest. Without puromycin treatment, no signal was detected, providing a clean background for targeted interactions (Supplementary Fig. 6A). We first performed CNOT3-Puro-PLA and demonstrated that CNOT3 is indeed present at the site of protein synthesis. The signals measured by both normalized cellular signal

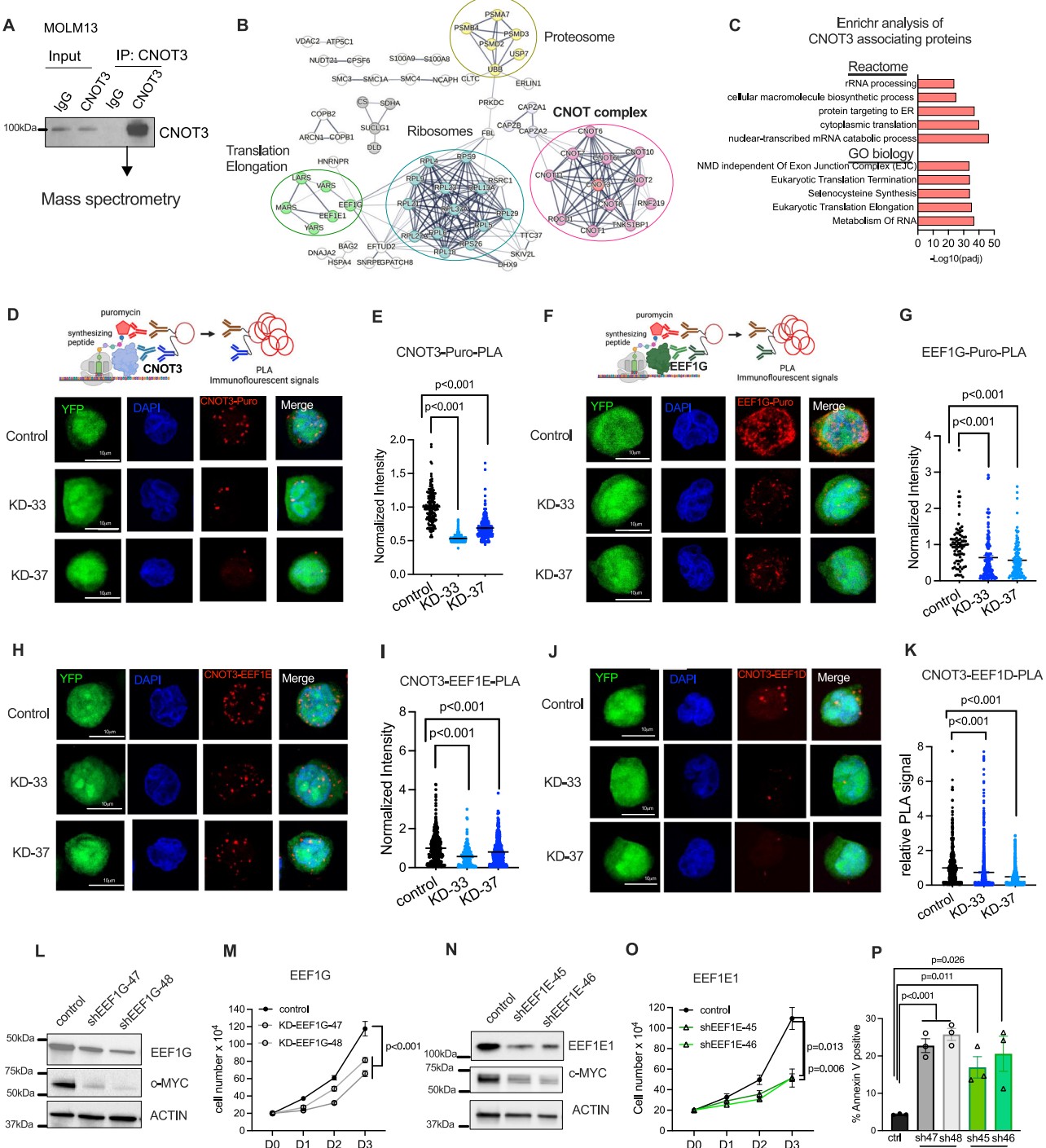

**Fig. 6 | CNOT3 associates with translation machinery in AML. A** Experimental scheme of immunoprecipitation and mass spectrometry analysis to uncover CNOT3 associating proteins. **B** STRING network analysis of proteins associating with CNOT3. **C** Enrichr analysis of functional protein groups preferentially associating with CNOT3 in leukemia cells. **D** Representative images of CNOT3-Puro-Proximity Ligation Assay (PLA) performed in control (scramble) shRNA vs. CNOT3 depleted KD-33 and KD-37 MOLM13 cells. **E** Quantitative summary of CNOT3-Puro-PLA in **D. F** Representative images of EEF1G-Puro-PLA performed in control (scramble) shRNA vs. CNOT3 depleted KD-33 and KD-37 MOLM13 cells. **G** Quantitative summary of EEF1G-Puro-PLA in **F**. Schemes in **D**, and **F** created using BioRender. Each dot represents data obtained from one cell. **H** Representative images of CNOT3-EEF1E-PLA performed in control (scramble) shRNA vs. CNOT3 depleted KD-33 and KD-37 MOLM13 cells. **I** Quantitative summary of CNOT3-EEF1E-

PLA in **H. J** Representative images of CNOT3-EEF1D-PLA performed in control (scramble) shRNA vs. CNOT3 depleted KD-33 and KD-37 MOLM13 cells. **K** Quantitative summary of CNOT3-EEF1D-PLA in **J**. All graphs show data as mean ± s.e.m. two-tailed Student's $t$ test, ***$p$ < 0.001. **L**–**P** MOLM-13 cells were transduced with lentiviruses expressing either a scramble (control) shRNA or EEF1E-targeting shRNAs (KD-47 and KD-48) or EEF1D-targeting shRNAs (KD-45 and KD-46). Cells were selected for puromycin resistance and assayed 3 days post-transduction. **L**–**N** Immunoblots showing efficient knockdown of EEF1E and EEF1D, respectively. ACTIN serves as loading control. **M**–**O** Cell proliferation. **P** Percentage of apoptotic cells by flow cytometry analysis for Annexin-V positivity. $n = 3$ independent experiments. All graphs showing data as mean ± s.e.m, $p$ values were calculated by two-tailed student's $t$ test. Source data are provided as Source Data files for figures (**A**, **C**, **E**, **G**, **I**, **K**–**P**).

intensity (Fig. 6D, E) and PLA foci intensity (Supplementary Fig. 6B) were significantly reduced when CNOT3 was knocked down, confirming the specificity of the interaction.

Next, we asked whether CNOT3 depletion influences EEFs' association with the translating proteins. EEFG1-Puro-PLA was performed in cells transduced with control vs. shRNAs against CNOT3. As expected for EEF1G, we observed strong PLA signals between EEF1G and puromycin, indicating abundant presence of EEF1G (Fig. 6F control panel). CNOT3 depletion resulted in substantial decrease in EEF1G-Puro-PLA (Fig. 6F, G and Supplementary Fig. 6C). These data strongly indicated that the presence of EEF1G at active translation sites is modulated by CNOT3, further solidifying the critical function of CNOT3 in regulating translation in leukemia.

We also implemented CNOT3-EEF1E-PLA and CNOT3-EEF1D-PLA and validated the interaction between CNOT3 and translation elongation factors (Fig. 6H−K). To further assess the requirement of the NOT box domain for the association of CNOT3 with the translation machinery, we used the CNOT3-FL and NOT box truncated overexpressing lines (described in Fig. 3) to perform PLA using Flag antibody to detect exogenous CNOT3 and antibodies against Puro, EEF1E and EEF1D. We observed that while PLA signals were strongly detected in CNOT3-FL overexpression, NOT box truncated CNOT3 overexpressing cells shown only a baseline detection equivalent to the signals observed in empty vector control cells (Supplementary Fig. 6D−I). These data strongly indicated that CNOT3 is associated with many components of the actively translating machinery. Given the association of CNOT3 with several translation elongation factors, we examined the functional relevance of these EEFs in leukemia cells. We efficiently knocked down the expression of EEF1G and EEF1E using two independent shRNAs for each gene in MOLM13 cells (Fig. 6L, N). Depletion of EEF1G and EEF1E both resulted in significant inhibition of cell growth (Fig. 6M, O and a strong induction of apoptosis in leukemia cells (Fig. 6P). We also observed a marked reduction of c-MYC expression in shRNA knocked-down cells (Fig. 6L, N). The results suggested that these associating proteins also play important roles in leukemia cell survival.

## Discussion

The CCR4-NOT complex, similar to ribosomes is generally perceived as common factors maintaining basic processes in cells. However, recent works have pointed to context-specific requirements for these regulatory mechanisms where fine-tunning levels of ribosomal proteins and tRNAs in coordination with codon optimality can influence gene expression programs favoring particular cancerous phenotypes[58,59]. Here, we demonstrated that the subunit CNOT3 of the CCR4-NOT complex is required for survival and growth of leukemia cells in vitro and in vivo. While the in vitro and in vivo phenotypes are highly agreeable, it remains to be determined whether loss of CNOT3 can lead to any homing defects in both normal and leukemia cells. The function is in contrast with previously characterized role of CNOT3 as a tumor suppressor in T-cell acute lymphoblastic leukemia[60]. While the observation is noteworthy, the different role of other proteins, such as SYNCRIP in AML[61] vs. T-ALL[62] has also been documented. Several missense mutations affecting Arg57 within the N-terminal coil-coil domain of CNOT3 were found in adult T-ALL patients. Interestingly, our results show the critical requirement of both the C-terminal NOT box domain and possibly part of the N-term domain in AML. Both regions are shown to be critically important for the CCR4-NOT complex to interact with and monitor ribosomes on mRNAs actively undergoing translation[36,63]. On the other hand, high expression of CNOT3 is associated with higher-grade colon cancer, exhibiting a similar oncogenic function[64]. In addition, we found that CNOT3 is differentially required in normal HSPCs vs. leukemia cells. In fact, loss of CNOT3 in cord blood-derived CD34+ cells resulted in rapid induction of differentiation, and sustained CNOT3 expression blocked

myeloid differentiation. It is known that CNOT3, CNOT1, and CNOT2 can maintain mouse and human embryonic stem cells (ES cells) by resisting differentiation toward the extraembryonic lineages[65,66]. These suggest that CNOT3 might play a more dominant role in the preservation of self-renewal and dormancy of stem cells. Altogether, these observations reinforce the notion of context-specific functions, hence the importance of defining the biological settings for corresponding phenotypic characterizations.

The molecular function of CNOT3 and the CCR4-NOT complex has been extensively studied in yeast and model organisms[35]. It has been shown to involve in many regulator steps of gene expression regulation – from transcription to mRNA export, deadenylation and mRNA decay, translation repression, codon sensing, and protein degradation[23]. However, in the mammalian system, the majority of previous studies linked the involvement of CNOT3's role in transcriptional control and mRNA degradation as the underlying mechanism for its function on various tissues such as liver[67], ES cells[66] and B cells[51]. It was a challenge to decouple the impact of CNOT3 and the CCR4-NOT deadenylation complex in regulating mRNA metabolism and translation activity. In this study, we presented a conclusive evaluation of CNOT3 functional interaction with c-MYC, a critically important oncogene in cancer. Genome-wide assessments of transcriptomic and proteomic changes upon CNOT3 knockdown showed a strong loss of expression of c-MYC downstream target genes and reduction in c-MYC protein abundance but not *c-MYC* mRNA itself. To account for contributions of all possible regulatory mechanisms, we then comprehensively examined all possible layers of regulation of *c-MYC* expression, from transcription to mRNA half-life, nucleo-cytoplasmic export, protein half-life and degradation and translation. *c-MYC* is a highly abundant and short-lived transcript[68] and its expression is tightly regulated at every single step[66,69]. Therefore, even an approximately twofold reduction of *c-MYC* within the highly translating ribosomal fraction can result in a significant decrease in its protein production as we observed upon CNOT3 depletion. It is noted that the assessments were performed at the earliest possible time point where we observed sufficient CNOT3 ablation. The particular experimental window allowed the status of protein expression vs. mRNA abundance to be captured, thereby enabling dissection of effects on mRNA metabolism vs. translation control.

The impact of CNOT3 loss on gene expression and translation is not global. Results obtained from both polysome profiling and RiboSTAMP assays clearly demonstrated that there is no gross effect on general protein synthesis. One of the major questions in the field of translation control is what factors modulate selective aspects of these generally considered promiscuous mechanisms. CNOT3 is not a direct RNA binding protein, and its association with RNA is often mediated by poly(A) binding proteins or other RNA binding proteins[35]. In order to gain insights into factors contributing to selectivity, we evaluated how each feature of an mRNA influence its abundance upon CNOT3 knockdown. A pattern emerged showing genes that exhibited low G/C and high A/T content and enriched some rare codons (i.e., lysine) to be preferentially promoted by CNOT3. In fact, transcripts with a high frequency of AU3 codons are shown to be translationally boosted to favor proliferation over differentiation[10,20]. Conversely, loss of CNOT3 increases the abundance of transcripts enriched for GC3. Assessment of translation activity using RiboSTAMP further substantiated the observation, demonstrating that loss of CNOT3 particularly decreased translation efficiency of actively translated transcripts encoding pro-proliferation proteins. It is postulated that reduced translation destabilizes the mRNA, thus, resulting in decrease in mRNA abundance. While there is solid data prescribing stable vs. nonstable codons in human cells[70,71], it is noted that optimal translation is also dictated by the pools of tRNAs and amino acids, which can be variable between tissues and cell states. Therefore, a conclusive statement on the coupling effects of mRNA decay and translation will require precise

measurements of these factors in each specific cell types. Overall, we propose that CNOT3 functions to boost translation efficiency, hence alleviating the bottlenecks in protein synthesis of growth-promoting proteins essential for the rapidly dividing leukemia cells.

The functions of a protein are highly dependent on its associating protein networks. CNOT3 is a conserved subunit of the CCR4-NOT complex where it makes stable interactions with the scaffolding subunit CNOT1 and subunit CNOT2[27,28]. It has been demonstrated that knockdown of CNOT3 results in reduced expression of other subunits of the complex including CNOT1 and CNOT2[72]. In our study, we observed minimal impact on CNOT1 protein abundance, at least at the early time points when CNOT3 was depleted in leukemia cells and HSPCs. While CNOT3 depletion might not have a strong impact on total protein abundance of other subunits in our system, it is possible that it might still impair assembly and stability of the complex, hence adversely affect functions of the entire complex.

On the other hand, bioID analysis using 293 T cells showed that CNOT3 and the CCR4-NOT complex mainly interact with cytoplasmic ribonucleoproteins, granule P-bodies, and the RISC complex to suppress translation activity[73]. In leukemia cells, we instead found a dominant association between CNOT3 (and the CCR4-NOT complex) and ribosomal proteins, tRNA synthetases, and translation elongation factors. The data aligns well with the transcriptomic and ribosome foot-printing analysis, supporting a role of CNOT3 in translation elongation. We further captured CNOT3 association with synthesizing peptides and demonstrated that the presence of elongation factors at the active translation sites is reduced in the absence of CNOT3. The direct involvement of CNOT3 in the translation process in mammalian cells has not been elucidated. In yeast, the study of Not5 (yeast ortholog of CNOT3) described direct binding of Not5 to the E sites of ribosomes lacking a tRNA at the A site, triggering disassembly of ribosomes on mRNAs and subsequently degradation of mRNAs[36]. This allows Not5 to perform surveillance of non-optimal decoding of ribosomes. A mechanism similar to what described in yeast has been shown to be employed in mammalian cells to couple translational control and mRNA stability[49]. However, it was not clear whether Not5 can also help resolving the slow-moving ribosomes prior to initiation of mRNA decay. Our data suggests that, at least in leukemia cells, CNOT3 plays a significant role in ensuring efficiency of translation process. Our model (Fig. 7) postulates that by associating with the translation machinery, CNOT3 acts to support synthesis of growth-promoting proteins while monitoring sub-optimal codons to moderate translation and mRNA stability. This regulatory mode will warrant timely turnover of inefficient processes and (re)distribution of resources to effectively translate mRNAs to optimally drive the malignant gene expression programs. Future studies are needed to further elucidate the functional link as well as the precise molecular basis connecting these regulatory processes.

In summary, we demonstrated that high level of translation efficiency mediated by CNOT3 is essential to sustain the expression of pro-growth and pro-survival genes during leukemogenesis. We uncovered CNOT3 as a key regulator of translation control in leukemia and nominated CNOT3 as a potential therapeutic target for the treatment of AML.

## Methods

The study complies with all relevant biosafety, animal procedures, and ethical regulations as approved by the University of British Columbia biosafety committee, animal care and use committee, and human Ethics Board. The study was conducted in accordance with the criteria set by the Declaration of Helsinki. All studies were performed according to animal protocol # A19-0101 approved by the Animal Care and Use Committee, and human ethics H19-02104 approved by the Research Ethic Board at the University of British Columbia.

### Cell culture

Human myeloid leukemia cell lines MOLM13 (AML French-American-British classification (FAB) M5a, t(9;11)); OCI-AML3 (hyperdiploid karyotype with NPM1 mutation type A and DNMT3A R882C); Kasumi-1 (hypodiploid karyotype, t(8;21) and KIT mutation N822); HL-60 (CCL-240); NB4 (APL/FAB M3; PML-RARA fusion); THP-1 (M5, t(9;11)) cells and MOLM-13-Cas9 cells (MOLM-13 cells engineered to constitutively express Cas9) and MV4-11-Cas9 (FAB M5, translocation t(4;11) and a *FLT3*-ITD mutation) were cultured in RPMI-1640 medium supplemented with 10% FBS, penicillin (100 units/ml), and streptomycin (100 units/ml). HEK293T cells were grown in DMEM medium with 10% FBS, penicillin (100 units/ml), and streptomycin (100 units/ml). Cells were incubated at 37 °C in a humidified 5% CO2 incubator. All cell lines were previously purchased from ATCC and validated and tested negative for mycoplasma contamination. All leukemia cell lines were authenticated via STR profiling using Genetica cell line testing service.

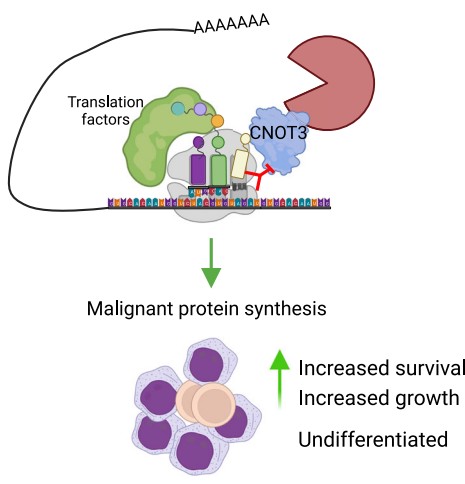
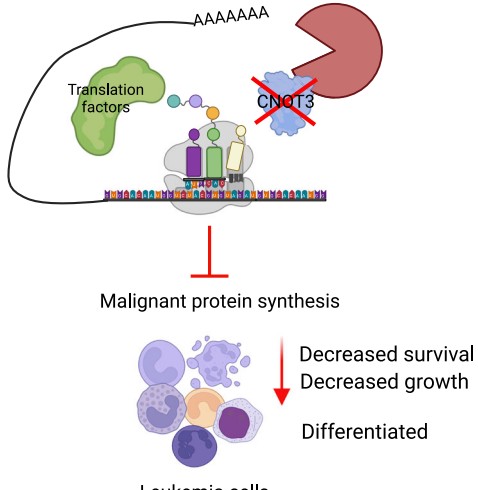

**Fig. 7 | Model for CNOT3 function in AML.** CNOT3 facilitates optimal translation by associating with translation machinery and monitoring decoding ribosomes to efficiently drive protein synthesis of malignant gene expression programs. This is critical for the maintenance of survival and the undifferentiated state of leukemia cells. Figure created using BioRender.

## Lentiviral plasmids

For shRNA-mediated depletion, pLKO.1 puromycin clones of shRNAs targeting *CNOT3* (KD-33-GCCCTGCCCTGGAAGACTGGA and KD-37-TGGAGCAGTTTGAAGATATTT) were obtained from MISSION™ shRNA Library (Sigma-Aldrich). YFP was swapped in to replace puromycin in the pLKO- shRNA-YFP constructs.

For CRISPR/Cas9- mediated deletion, sgRNAs specifically target *CNOT3* (sgRNA#1-TGAGGGACCAAATCAAGACA and sgRNA#2-GTGGCCCCACCAGCTCCCAG) are cloned into pLKO5.sgRNA.EFS.GFP. sgRNAs were chosen from the human sgRNA library used in Wang et al.[40]

For overexpression of *CNOT3* and NOT box truncated *CNOT3*, full-length *CNOT3* cDNA was synthesized by Azenta Life Sciences (previously Genewiz). The cDNA and truncated fragment were subcloned into pMNDU3-PGK-YFP lentiviral backbone.

## Virus production

293T cells were seeded in 10 cm plates and grown to 70-80% confluency before cell transfection. The lentiviral transfer vector plasmid, packaging plasmid (psPAX2), and envelop plasmid (pMD2.G) were combined at 4:3:1 ratio (12.5 µg:9.375 µg:3.125 µg, respectively) in 1 mL of 0.25 M CaCl2. The mix of plasmid DNA was added into 1 mL of filtered 2× concentrate BES buffered saline (Millipore 14280, bioWORLD 40220006-2) in a dropwise manner. A sterile 1 mL pipette was used to gently create bubble air through the DNA mix. Following the incubation at room temperature for 5–10 mins. the solution was added dropwise to 293 T cells. The plates were rocked gently in a circular motion to distribute the precipitates. Cells were cultured at 37 °C in a humidified 5% CO2 incubator for 16 hours, followed by replacement with fresh media. Viral supernatants were collected twice at 24 hours and 48 hours after replenishing new media. The supernatant was passed through a 0.45 µm pore PVDF Millex-HV filter (Millipore, SLHVR33RB) and stored at −80 °C.

## Viral transduction and cell selection

Leukemia cells were transduced with lentiviruses by spinfection at $2600 \times g$, room temperature for 1 hour. Cells were plated in RPMI with 10% FBS at the density of $0.5 \times 10^6$ cells/ml.

For puromycin selection, 3 µg/ml puromycin was added into transduced cells at 24 hours after the transduction procedure. Cells were collected at ~48–56 hours post transduction after ~24–32 hours of puromycin selection.

For selection using GFP and YFP reporters, transduced cells were replenished with new media at 24 h after transduction. At 72 hours post viral transduction, cells were sorted based on GFP or YFP positivity.

Selected cells were then either plated at $0.2 \times 10^6$ cells/ml for proliferation assays or washed and collected for downstream analyses.

## Domain screening

The entire coding sequence of *CNOT3* was scanned for the design of sgRNAs. In total, 464 sgRNAs tiling along the whole region were selected. A pool of CNOT3-targeted sgRNAs together with control sgRNAs targeting safe harbors or non-targeting sgRNAs was generated. Viral titration was performed to determine the multiplicity of infection (MOI) to be ~0.3. Viruses were then transduced into MOLM13-Cas9 cells. Transduced cells were selected with puromycin for 72 hours prior to D0 collection and plated to grow for an additional 14 days before D14 collection. DNA from cells collected at the two-time points was extracted and sequenced to determine the representation of each sgRNA before and after the proliferation period. CRISPR scores were determined by fold enrichment of sgRNAs and further processed by CRISPRO program[48] via a LOESS (LOcally WEighted Scatter-plot Smoother) regression model to calculate local trends of the perturbation effect and generate final annotated results.

## Culture and lentiviral transduction of primary normal and leukemia cells

Cord blood-derived CD34+ cells (CB-CD34+ cells) and primary AML patient samples were obtained from the hematology cell bank at BC Cancer Research Center. Cryopreserved samples were thawed in thawing media containing 4.5 mL IMDM, 4.5 mL FBS, and 1% DNase I (10 mg/ml).

Cells were cultured in 96 well plate (100,00 cells in 100 µl media/well) and in serum-free media (80% IMDM, 20% BIT, 2-Mercaptoethanol (100 nM) and L-Glutamine (2 mM)) supplemented with 5 growth factors (IL3 (PeproTech, 200-03-50UG, 20 ng/ml), IL6 (PeproTech, 200-06-50UG, 20 ng/ml), SCF (PeproTech, 300-07-100UG, 100 ng/ml), G-CSF (PeproTech, 300-23-50UG, 20 ng/ml), hFLT3-ligand (PeproTech, 300-19-50UG, 100 ng/ml)) and UM171 (35 nM) and SR1 (750 nM) [StemCell Technologies, 72354]. The cells were incubated at 37 °C in humidified 5% CO2 incubator.

Prior to viral transduction, cells were seeded for 6–12 hours. 10 µl of concentrated viruses (titers ~10⁹/ml) was added per well. Plates were then spined at $2600 \times g$, for 1 hour at 37 °C. The cells were incubated with viruses for 6 hours before the media was changed to fresh media.

For CB-CD34+ cells, puromycin was performed by adding 1 µg/ml puromycin at 24 hours post-transduction, and cells were collected after 48 hours selection period. Isolation of transduced YFP cells was performed by flow cytometry at 72 hours post-transduction. Selected cells were then either plated in basic media (SFM (80% IMDM and 20% BIT) supplemented with human (h) SCF (100 ng/ml), hTPO (100 ng/ml), hFLT3-ligand (100 ng/mL), and IL6 (20 ng/ml)) for proliferation or myeloid promoting media (SFEM supplemented with SCF (100 ng/ml), hFLT3-ligand (10 ng/ml), IL-3 (20 ng/ml), IL-6 (20 ng/ml), GM-CSF (PeproTech, 300-03-20UG, 20 ng/ml) and G-CSF (20 ng/ml)) to induce differentiation.

For primary AML patient cells, selection of transduced cells was based on sorting of YFP-positive cells at 48 hours post-transduction. Collected cells were then subjected to downstream assays, including liquid culture, colony-forming assay, and in vivo transplantation.

## Colony-forming cell (CFC) assay

CFC assays were performed in a semi-solid methylcellulose medium by using MethoCult H4100 (StemCell Technologies, 04434). 1000 primary leukemia cells and 1500 CB-CD34+ cells were plated in six-well plate containing 1 ml methylcellulose. Plates were kept in incubator at 37 °C and 5% CO2. The colonies were scored under a microscope after 14 days.

## Morphological analysis

$2 \times 10^5$ cells were centrifuged onto slides for 5 minutes at $860 \times g$ and air-dried prior to GIEMSA staining. The cell morphology was evaluated by Nikon A1-si Confocal TIRF Microscope.

## Flow cytometry

To monitor cellular differentiation status, cells were stained with the following antibodies: APC-CD11b (ThermoFisher, CD11B05), FITC-CD13 (ThermoFisher, 11-0138-42) and PE-CD14 (ThermoFisher, 12-0149-42). To measure apoptosis, cells were washed with PBS and incubated with APC-Annexin V (BD Biosciences, 550475) in the annexin-V binding buffer in a reaction volume of 100 µl for 15 minutes according to the manufacturer's instructions. DAPI was added prior to analysis. Cells were analyzed on a BD FACS LSR Fortessa instrument.

To assess engraftment of human leukemia cells in NSG/NSG-3GS recipient mice, blood and bone marrow cells collected from animals were treated with red blood cell (RBC) lysis buffer [NEB, 46232 S] prior to staining with two antibodies against human CD45 surface marker i.e., anti-hCD45-AF700 clone 2D1 (Biolegend, 368514) and anti-hCD45-PE clone H130 (eBioscience,12-0459-42). All flow cytometry samples were analyzed on a BD FACS LSR Fortessa instrument.

For CNOT3 intracellular flow cytometry analysis (IC flow), cells were washed with PBS and resuspended in 250 µl of 1.5% paraformaldehyde as fixative for 15 minutes in RT. After washing the pellet with PBS, 1 ml cold methanol was added for 20 minutes to permeabilize the cells. The cells were washed and 5 µl of primary intracellular CNOT3 antibody and incubated for 30 minutes at 4 °C. Cells were washed and stained with anti-mouse secondary antibody conjugated with fluorochrome AF488 (Invitrogen, A-21202), and incubated at room temperature for 30 minutes. The cells were resuspended in 200 µl of FACS buffer, transferred to FACS tube and were analyzed.

### In vivo transplantation of AML cells and patient-derived xenografts

For AML cell lines i.e., MOLM13 and OCI-AML3, $0.5 \times 10^6$ cells were tail-vein injected into NSG (NOD-scid IL2Rgnull) mice. For primary AML patient cells, $0.3–0.5 \times 10^6$ cells were tail-vein injected into NRG-3GS (NOD.Rag1−/−;γcnull-IL3/GM/SF) mice. Recipient mice were sub-lethally irradiated (800 cGy) one day prior to injection. NSG and NSG-3GS recipients were all 8–10-week-old female mice. All mice were purchased from BCCRC mouse facility, and procedures all complied with Animal protocols approved by the University of British Columbia.

### RNA extraction and quantitative real-time PCR assays (qRT-PCR)

Total RNA was extracted from cells using RNeasy Plus Mini Kit (QIAGEN) following the standard manual. For qRT-PCR, an equal amount of RNA from samples was reverse transcribed into cDNA with iScript™ Reverse Transcription Supermix (Biorad, 1708841), and qPCR was performed using a QuantStudio™ 5 Real-Time PCR System detection using primers together with SYBR green master mix (ABI systems). All reactions were run in triplicate in three independent experiments and amplified in a 10 µl reaction according to the manufacturer's protocol. Cycling conditions included an initial hold step (95 °C for 10 seconds) and 40 cycles of a two-step PCR (95 °C for 15 seconds and then 60 °C for 60 seconds), followed by a dissociation step (95 °C for 15 seconds). Relative messenger RNA (mRNA) expression was calculated by the comparative $2^{-\Delta\Delta CT}$ method. *ACTIN* was used as an internal control.

### RNA stability assay

Cells were plated at $0.5 \times 10^6$ cells/ml and treated with 5 µM Actinomycin D for inhibition of mRNA transcription. Cells were collected at 0 minute, 15 minutes, 30 minutes, 60 minutes, 2 hours and 4 hours post treatment, and total RNA was extracted and used for qRT-PCR.

### Protein half-life and degradation assay

Cells were plated at $0.5 \times 10^6$ cells/ml and treated with CHX to inhibit protein synthesis and MG132 to suppress activity of proteosome. Cells were collected at indicated time points after treatment, and total proteins were extracted and used for immunoblot analysis.

### Immunoblot analysis

For immunoblot analysis, cells were counted and washed twice with cold PBS prior to collection. 200,000 cells were lysed in 40 µL 1× Laemmli protein loading buffer and boiled at 95 °C for 5 minutes. Whole-cell lysates were run on a 4%–15% TGX precast protein gels or 4–15% Criterion™ TGX™ Precast Midi Protein Gel (Biorad,4561085, 5671083, 5671084) and transferred to a nitrocellulose membrane (Biorad, 1620115). Membranes were blocked with 5% milk for 1 hour prior to incubation with antibodies against CNOT3 (Abnova, H00004849-M01, 1:1000), c-MYC [NEB, 5605 S, 1:1000], p21 [CST, 2947 S, 1:1000], ACTIN (Sigma Aldrich, A3854, 1:5000). After overnight primary antibody incubation, the membranes were washed with 1 × PBST and then incubated with HRP-linked secondary antibodies (goat anti-mouse IgG [NEB, 7076 V] or goat anti-rabbit IgG [NEB, 7074 V], 1:3000) for 1 hour at room temperature. Immobilon ECL Western HRP Substrate (Millipore, WBKLS0500) and ECL Western

Blotting Detection Reagent (Sigma, GERPN2105) were used to detect the protein bands on the Bio-Rad ChemiDoc Imaging System with chemiluminescence detection. The relative protein expression of each protein band was analyzed by Image J software, using β-actin as the reference. Uncropped and unprocessed scans of the most important blots were included in the Source Data file.

### Immunofluorescence and RNA-FISH

Cells were fixed in 10% neutral buffered formalin at a concentration of $10^6$ cells/mL for 60 minutes at 37 °C. They are then resuspended in 70% Ethanol, and $0.15–0.2 \times 10^6$ cells are spun onto slides at $1300 \times g$ for 5 minutes using the ThermoFisher Cytospin 4. Cells are then permeabilized with 50%, 70%, and $2 \times 100\%$ Ethanol for 5 minutes each. After outlining the cell spot with a hydrophobic pen, cells are incubated in 0.5% PBST blocking solution for 60 minutes at room temperature. Cells were then incubated in *c-myc* FISH probe (ACD bio-techne, 311761-C3) for 120 minutes at 40 °C. Consequent FISH signal amplification steps were performed according to the protocol outlined in the RNAScope Assay kit from ACD bio-techne. Primary antibodies were prepared in the blocking solution mouse α-CNOT3: 10 µg/mL (Abnova, H00004849-M01); Cells were incubated in primary antibodies overnight at 4 °C. The cells were then washed and incubated with secondary antibodies (Invitrogen, anti-mouse AF488: 1:500 (A-21202), anti-mouse AF568: 1:100 (A10037); anti-rabbit AF568: 1:500 (A10042) and anti-rabbit AF647: 1:500 (A32795)) for 60 minutes at room temperature. Finally, cells were washed again and mounted in a MOWIOL mounting medium. Slides were stored at −20 °C for at least 24 hours before imaging on the Zeiss LSM 800 AiryScan Confocal Microscope. Images were analyzed using ImageJ (Fiji) and statistically analyzed on GraphPad Prism.

### Puromycin-proximity labeling assay (Puro-PLA)

For Puromycin-PLA, cells were incubated in 2 µM Puromycin culture media at a concentration of $10^6$ cells/mL for 30 minutes before fixation. Cells were fixed in 10% neutral buffered formalin at a concentration of $10^6$ cells/mL for 60 minutes at 37 °C. They are then resuspended in 70% Ethanol, and 150–200k cells are spun onto slides at $1300 \times g$ for 5 minutes using the ThermoFisher Cytospin 4. Cells are then permeabilized with 50%, 70%, and $2 \times 100\%$ Ethanol for 5 minutes each. Cells are incubated in the blocking solution provided with the Duolink® In Situ Red PLA Kit at 37 °C for 60 minutes. Primary anitbodies (Mouse/Rabbit combinations were chosen for compatibility with the PLA kit) were prepared in the antibody diluent solution provided with the kit (Rabbit α-puromycin: 1:50 (Abclonal, A23031); mouse α-EEF1G: 5 µg/mL (Novus, 3F11-1A10); mouse α-CNOT3: 10 µg/mL (Abnova, H00004849-M01) were incubated in primary antibodies overnight at 4 °C. PLA probe treatment, Ligation and Amplification steps were performed according to the protocol provided by manufacturer. Cells were finally mounted in medium containing DAPI provided with the kit and imaged a day later on the Zeiss LSM 800 AiryScan Confocal Microscope. Images were analyzed using ImageJ (Fiji) and statistically analyzed on GraphPad Prism.

### Polysome profiling

Polysome profiling was adapted from the protocol developed by Dr. Chris Hughes and was performed with assistance of Dr. Hughes in the laboratory of Dr. Sorensen at BC Cancer. MOLM13 cells (transduced with shRNAs – control vs. KD-33 and KD-37) were collected by spinning down at $2600 \times g$ for 5 minutes at 4 °C, washed twice with PBS, and then treated with 100ug/ml cycloheximide (CHX) for 10 minutes at 37 °C. After centrifuge, the supernatant was aspirated and the pellet were lysed and loaded on top of a 15–40% sucrose gradient. After ultracentrifugation, samples were loaded onto the BioComp Gradient Fractionator (GST, BioComp). RNA concentration was monitored and recorded at A254 for each fraction. The results were used to plot global

profiles showing distribution of fractions containing free RNA, ribosomal subunits, mono-disomes and polysomes. Based on the profiles, fractions covering monosomes vs. polysomes were combined and used for RNA extraction using Zymo RNA extraction kit [Zymo Research, R2061]. The abundance of mRNAs was analyzed by RT-qPCR. Actin was used as control.

## Co-immunoprecipitation and mass spectrometry analysis
MOLM13 cells were collected by spinning down at $2600 \times g$ for 5 minutes at 4 °C, washed twice with cold PBS, and then resuspended thoroughly at $2 \times 10^7$ cells in 1 ml of 1× RIPA lysis buffer (ThermoFisher, 89901) with freshly added DTT (1 mM) and Halt Protease Inhibitor Cocktail (ThermoFisher, 78430). The cells were incubated for 30 minutes on ice. Supernatant was then collected after the mix was spun at $24,104 \times g$ for 30 minutes at 4 °C. For each immunoprecipitation assay, 250 µl of cell lysate was mixed with 750 µl of 1× Ripa buffer, 10 µg of anti-CNOT3 antibody or 10 µg of anti-mouse IgG antibody, and 50 µl pre-washed Dynabeads protein A/G magnetic beads (Invitrogen, 10015D). The mixture was rotated overnight at 4 °C. After immunoprecipitation, protein lysates were removed and beads were washed 6 times with the binding 1× RIPA lysis buffer and 2 times with PBS. Proteins were then eluted from Dynabeads using 8 M Urea (GE Healthcare Life Sciences), reduced, alkylated, and digested with Endopeptidase Lys-C (>4 M Urea) for 6 hours followed by overnight trypsinization (>2 M Urea). Peptides were analyzed by reversed phase nano LC-MS/MS (Ultimate 3000 nano-HPLC system coupled to a Q-Exactive Plusor a Fusion Lumos mass spectrometer, operated in high/low mode (ThermoFisher) at the Proteomics Resource Center at Rockefeller University. Identified peptides were filtered using 1% False Discovery Rate (FDR) and Percolator. Proteins were sorted out according to the highest area. The area is calculated based on the 3 most abundant peptides for the respective protein. Proteins not detected or present in low amounts are assigned an area zero. Data were extracted and queried against Uniprot human database using Proteome Discoverer and Mascot.

## TMT-mass spectrometry for proteomic profiling
Harvested cell pellets were thawed on ice prior to lysis. To each pellet, 200 µL of lysis buffer (50 mM HEPES pH 8 (Sigma, H3375), 4 M guanidine hydrochloride (Sigma, G4505), 10 mM tris (2-carboxyethyl) phosphine hydrochloride (Sigma, C4706), 40 mM chloroacetamide (Sigma, C0267), and 1× cOmplete protease inhibitor – EDTA free (Sigma, 4693159001) in HPLC grade water (Sigma, 270733) was added. Lysis mixtures were pipette mixed and then transferred to 2 mL FastPrep-compatible tubes containing Lysing Matrix D (MP Biomedicals, 116913050). Lysis mixtures were vortexed on the FastPrep-24 5 G instrument (6 M/s, 45 seconds, 2 cycles). Tubes were centrifuged at $20,000 \times g$ for 1 minute, and the supernatant recovered. Resultant lysates were heated at 95 °C for 15 mins, and cooled to room temperature. Protein concentrations were measured using the Pierce BCA Protein Assay Kit (ThermoFisher,23225). Lysates were clean up prior to TMT labeling using TMT 10- and 6-plex labeling kits (Pierce). Labeled peptides were concentrated in a SpeedVac centrifuge to reduce acetonitrile, and the differentially labeled peptides were combined. Clean up using SepPak C18, 50 mg columns (WAT054960) prior to HPLC fractionation. High-pH reversed phase separation was performed on an Agilent 1100 HPLC system equipped with a diode array detector (254, 260, and 280 nm). Fractionation was performed on a Kinetix EVO C18 column (2.1 × 150 mm, 1.7 µm core shell, 100 Å, Phenomenex). Elution was performed at a flow rate of 0.25 mL/min using a gradient of mobile phase A (10 mM ammonium bicarbonate, pH 8) and B (acetonitrile), from 3% to 35% over 60 minutes. Fractions were collected every minute across the elution window for a total of 48 fractions, which were concatenated to 12 final fractions (e.g.,

1 + 13 + 25 + 37 = fraction 1). Fractions were dried in a SpeedVac centrifuge and reconstituted in 0.1% formic acid HPLC water before subjected to MS analysis. Analysis of peptide fractions was carried out on an Orbitrap Fusion Tribrid MS platform (ThermoFisher). Data was processed using Sequest in Proteome Discoverer version 2.4 against the human fasta database downloaded from Uniprot in November 2019 along with the CRAPome[74]. Percolator was used to determine the FDR, and only peptides with an FDR < 0.01 were considered during analysis. Data analysis was performed using a PECA analysis in R version 3.6.1.

## RiboSTAMP
A lentiviral expressing plasmid was a gift from Dr. Nika Shakiba at the University of British Columbia. The plasmid expresses RPS2 proteins fused with APOBEC1 with HA tag at the C-terminus under inducible TRE promoter. mRuby was also expressed following the HA tag under the same promoter with P2A ribosomal skip site. A stable MOLM13 cell line expressing a doxycycline-inducible RPS2 small ribosomal subunit tagged with APOBEC1 protein was generated with lentiviral transduction. Inducible cells were seeded at $0.5 \times 10^6$ cells/ml in six-well plates. Control scrambled or CNOT3-targeting shRNAs were transduced into the inducible cells with lentiviral along with 5 µg/ml doxycycline and 5 µg/ml polybrene. Transduced cells were selected with fresh media containing 3 µg/ml puromycin and 5 ug/ml doxycycline at 16 hours post-transduction. Cells were collected at 56 hours post-transduction for immunoblot analysis of HA tag and CNOT3 to validate the expression of fusion protein and knockdown efficiency and RNA extraction for RNA sequencing.

## RNA-sequencing for transcriptomic profiling
Total RNA isolated from biological samples was processed with Poly(A) selection and library preparation using TruSeq RNA Library Prep Kit v2 (Illumina, RS-122-2001). Libraries were sequenced with 2 × 150 bp paired-end run and, in total 20–40 million reads/sample using Illumina® NovaSeq.

## Bioinformatic analysis and gradient boosting
Cutadapt (v1.18)was used to trim reads, and RSEM (v1.3.1) (2) was to align to either the human (Ensembl v38). Differential expression analysis was carried out using DEseq2 v1.40.2 (3) with lfcShrink (type = "apeglm") (4) to account for lowly expressed mRNAs. Differential expression analysis, visualizations, and statistics used R version 4.3.1. In cases of multiple comparisons, 1-Way ANOVAs were conducted, followed by a post-hoc Tukey test. For instances in which the assumptions of ANOVA were violated, data was either log-transformed or the non-parametric Kruskal–Wallis test was used followed by a paired Wilcoxon test. All shell scripts and R scripts can be found at the following url: https://github.com/JamesEttles/Translation-efficiency-driven-by-CNOT3-subunit-of-the-CCR4-NOT-complex-promotes-leukemogenesis/tree/main.

Gradient Boosting: Data was filtered for significantly differentially expressed genes (Benjamini and Hochberg method; $p$.adj <0.05), and the properties of the most abundant transcript per gene, as determined from RSEM analysis, were annotated with their respective features. Correlated features were removed (Pearson correlation coefficient >0.75) prior to model training, and lengths of 5′UTRs, CDSs, and 3′UTRs were log-transformed. Numeric variables were normalized and categorical variables were one-hot encoded. Data was then split at random into training and test datasets at a 70:30 split, respectively. Model performance was evaluated using receiver operator curves and area under the curve metrics and assessed for overfitting. The relative contribution of features in determining model output was interrogated using the vip package. Hyperparameters used for the model used in each dataset are included in the supplementary table in Supplementary information.

## RiboSTAMP -sequencing data analysis

Sequencing paired-end reads were processed using FASTQ/fastp (v0.23.4)[75] for adapter trimming, quality filtering, and quality control. Reads were then aligned to hg38 v109 genomes download from Ensemble (Homo_sapiens.GRCh38.dna.primary_assembly.fa.gz) using STAR (v2.7.10b)[76]. Raw read counts were created using Subread featureCounts (v2.0.3)[77].

RiboSTAMP analysis was performed as previously described[54]. In brief, C-to-U edits are identified as C-A mutations and quantified using the SAILOR (v1.2.0) analysis pipeline. C-to-U mismatches were scored using a beta distribution that factors both site coverage and editing percentage following the removal of annotated single-nucleotide polymorphism (SNPs) dbSNP155 downloaded from UCSC Genome Browser website to generate confidence score. Sites with confidence score ≥0.5, and found in at least two replications were carried to downstream analysis. The value nsites is the total number of edit sites scored for each gene and within the coding region (CDS). Edits per kilobase of transcript per million mapped reads (EPKM) per gene were calculated by summing up CDS (as defined by hg38 v109 Gencode annotations) edit counts per gene and divided by "per million" mapped read counts to CDS, respectively, for all genes with read counts greater than 0. This number was then normalized to the length of either the CDS of each gene in kilobases (kb). MetaPlotR was used to generate the metagene plot to show edit (≥0.5 confidence score) distribution. We converted the transcriptomic coordinates to metagene coordinates such that sites that occur in the 5'UTR have a value from 0 to 1, where 0 and 1 represent the 5' and 3' ends of the 5'UTR, respectively. Similarly, sites in the CDS have a value from 1 to 2 and the 3'UTR 2 to 3. For differential transcriptomic gene expression DESeq2 (v1.2.10; default parameters) was performed, and for differential EPKM level comparison, limma (v 3.56.2) was performed.

## Survival analysis

Survival analysis was performed on TCGA-LAML[43] and BEATAML[44] cohorts. FPKM normalized or Harmonized CNOT3 counts were extracted from each dataset, respectively. Cutoffs to separate patients into high and low CNOT3 expressions were calculated using Maximally Selected Rank Statistics. Kaplan–Meier survival analysis was then performed using these cutoffs. p values were calculated using the log-rank test.

## Statistics and reproducibility

Statistical tests were defined for each dataset. No statistical method was used to predetermine sample size. No data were excluded from the analyses. We allocated recipient mice into different groups randomly in transplant in vivo experiments. Animals in all experiment groups are sex and age-matched. No other randomization was performed in the study. The investigators were not blinded to outcome assessments. All experiments were performed to have at least three biological replicates to ensure power for statistical analysis using two-sided student t test. Student's t test was used when the two-sample equal-variance model with normal distribution was fitted. When the variances of the two groups are unequal i.e., the sample sizes or the measurement variability differ between the groups, Welch's t test was used. When the data are not normally distributed, or the variances are unequal, Wilcoxon rank-sum test was used to compare the ranks within the dataset. Sample groups for all experiments were not blinded. p values equal to or <0.05 were considered to be significant. All immunoblots are performed at least n = 3 biological replicates unless specified in figure legends. Graphs and error bars reflect means + s.e.m., except where stated otherwise. For animal studies, survival probabilities were estimated using the Kaplan–Meier method and compared with the log-rank (Mantel-Cox) test. All animals were randomly assigned to the experimental groups. All statistical analyses were carried out using GraphPad Prism 4.0 and the R statistical environment.

## Reporting summary

Further information on research design is available in the Nature Portfolio Reporting Summary linked to this article.

## Data availability

The data supporting the findings of this study are available from the corresponding authors upon request. Raw and assembled sequencing data generated in this study have been deposited in NCBI's Gene Expression Omnibus (GEO) under accession BioProject PRJNA985375. Source data are provided as Source Data file. Source data are provided with this paper.

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

## Acknowledgements

The work was supported by the Canadian Institutes for Health Research to L.P.V. (AWD-020547) and Natural Sciences and Engineering Research Council of Canada Discovery Grant. L.P.V. is a Special Scholar of the Leukemia and Lymphoma Society, a Scholar of the American Society of Hematology, and is a Tier 2 Canada Research in RNA biology and Hematological Malignancies and is supported by the Michael Smith Health Research Scholar award. L.P.V.'s lab is also supported by the Terry Fox Research Institute New Investigator Award. M.B., J.A.W., and J.E. were supported by Cancer Research UK core funding to the CRUK Beatson Institute (A31287) for M.B.'s lab (A29252) and to the CRUK Scotland Center (CTRQQR-2021\100006). We appreciate Dr. Nika Shakiba (University of British Columbia) for sharing with us the RiboSTAMP plasmid. We would like to acknowledge the patients and their families who contributed to this study via the hematology cell bank at the University of British Columbia. We thank the staff of the British Columbia Cancer Center Flow Core and Animal Research Center for their technical assistance. We are grateful to our lab and Eaves lab members for their discussion and technical assistance. The results shown here are in part based upon data generated by the TCGA Research Network: https://www.cancer.gov/tcga and the Beat AML program—a project supported by the Leukemia & Lymphoma Society and the OHSU Knight Cancer Institute.

## Author contributions

M.G.H. designed, performed experiments, analyzed data and wrote sections of the methods within the manuscript. Y.L. designed and performed experiments analyzed data, and wrote the manuscript. J.E. performed gradient boosting analysis and wrote the corresponding results and methods; G.B. performed intracellular staining of CNOT3 and provided data on primary patient samples. N.V. and Z.J. performed and analyzed immunofluorescent. Z.L. performed Ribo-stamp analysis and wrote the corresponding method section. L.E. performed survival analysis and wrote the corresponding method section. S.S.M. performed TMT-mass spectrometry for proteomic profiling and wrote the corresponding method section. Y.K., J.A.W., K.D, K.M., K.Y., T.T, M.Y., A.A., and G.E. provided critical technical support and performed experiments. A.K., G.M., F.K., and F.B. provided supervision and critical suggestions. M.B. provided supervision for bioinformatic and gradient boosting analysis supervision and critical suggestions. L.P.V. obtained funding, led and directed the project, designed and performed experiments, analyzed data, and wrote the manuscript.

## Competing interests

The authors declare no competing interest.
