## [Peer Review File · Nature Communications]

Translation efficiency driven by CNOT3 subunit of the CCR4-NOT complex promotes leukemogenesisREVIEWER COMMENTS

Reviewer #1 (Remarks to the Author):

Review of Manuscript NCOMMS-23-41711-T by Dr. Vu and Co-Authors Titled "Translation Efficiency Driven by the CNOT3 Subunit of the CCR4-NOT Complex Promotes Leukemogenesis"

In this study, Dr. Vu and their co-authors have presented compelling evidence regarding the pivotal role of CNOT3 in enhancing the translation of c-MYC, thereby promoting leukemogenesis. The authors skillfully demonstrated that the expression of the CNOT3 subunit within the CCR4-NOT complex is significantly upregulated in Acute Myeloid Leukemia (AML). They convincingly showed that CNOT3 actively fosters the survival and proliferation of leukemia cells, as effectively depicted in Figure 1. Furthermore, the authors established that CNOT3 is indispensable for leukemogenesis and the maintenance of undifferentiated states within human hematopoietic stem/progenitor cells (HSPCs), as visually presented in Figure 2. Notably, they identified the NOT box domain as a crucial component for CNOT3 function in leukemia cells, as highlighted in Figure 3. Additionally, the authors provided valuable insights into how CNOT3 regulates c-MYC expression at the translational level, as exemplified in Figure 4. They also conducted a comprehensive global assessment of CNOT3 mRNA targets, as demonstrated in Figure 5.

The manuscript's exploration of the unique and intriguing translation regulation of c-MYC by the major deadenylase CCR4-NOT is of significant interest to the readership of Nature Communications. However, further mechanistic insights are warranted. I recommend the publication of this study in Nature Communications, contingent upon the incorporation of additional validation of CNOT3's function in c-MYC translation. I would like to raise a couple of important points for consideration:

While the study has made significant strides in uncovering CNOT3's role in c-MYC translation, its mechanistic insights remain elusive. Given the essential role of the NOT box domain in the CNOT3 function (Figure 3D), it is imperative to conduct polysome analysis of c-MYC in the context of NOT box-truncated CNOT3, as illustrated in Figure 4H. Furthermore, investigating the association of the NOT box-truncated CNOT3 with the translation machinery, as outlined in Figure 6, is also crucial to provide additional valuable insights.

Although the authors have convincingly demonstrated the essential role of the NOT box domain in CNOT3 function in AML (Figure 3D), it is advisable to explore whether the N-terminal domain of CNOT3 also plays a crucial role in its function and its impact on c-MYC translation in AML. I strongly encourage the authors to expand their investigation of c-MYC translation (Figure 4) and the global assessment of CNOT3 (Figure 5). This additional analysis would contribute to a more comprehensive understanding of the mechanisms underlying CNOT3-mediated translation regulation.

Reviewer #2 (Remarks to the Author):

In this manuscript Ghashghaei et al. dissect the role of CNOT3 and the CCR4-NOT complex in hematopoiesis and leukemogenesis. They identify CNOT3 as overexpressed in 3 AML datasets which peaks their interest in this protein and whether it may specifically contribute to leukemia survival. They test several AML cell lines, all of which express CNOT3 protein with higher expression in some than in normal human cord blood hematopoietic stem and progenitor CD34+ cells. They show that knockdown of CNOT3 via shRNA and deletion via CRISPR results in reduced AML growth using various AML models. Knockdown in normal CD34+ results in lack of differentiation and reduced colony formation. Overexpression has the opposite effects.

Next they perform CRISPR tiling to identify the domains of CNOT3 responsible for this effect. They identify N- and C-terminal domains of interest, but focus subsequently on the C-terminal NOT box domain. They confirm functionally, that the NOT box domain is essential for CNOT3's function.

They subsequently perform transcription and proteomic profiling of control and CNOT3 knockdown leukemia cells and identify c-MYC as CNOT target. Various techniques serve to dissect the mechanism of this regulation, suggesting the c-MYC translation is affected by CNOT3 KD.

Given that CNOT3 participates in the CCR4-NOT complex mediated translational control, they next analyzed transcripts that were differentially regulated for nucleotide usage and codon composition. They suggest that CNOT3 regulates expression of genes based on bias at the third "wobble" codon that correspond to the cell's proliferative state. Via Ribo-STAMP they also analyze the translational efficiency in wildtype and CNOT3 knockdown cells suggesting that CNOT3 may enhance translation of highly translated mRNAs. Indeed, via CNOT3 Co-IP followed by mass spec and validation via the Puro-PLA assay, they validate association of CNOT3 with the translational machinery.

Overall, this is a highly informative and very well executed study.

The authors very thoroughly dissect the role of CNOT3 in leukemogenesis and carefully analyze its mechanistic role in mammalian cells. The data is very well presented.

Two conceptual questions arise:

1) The authors suggest that CNOT3's role is unique to AML and that its targeting could represent a therapeutic opportunity. They do not determine how CNOT3 and CCR4-NOT machinery is upregulated in AML. This should be discussed. The data on CD34+ cells is somewhat limited, restricted to umbilical cord blood (instead of adult CD34+) and liquid culture and CFU assays without serial replating. The latter should be performed to assess not only progenitors but more immature cells. While engraftment of HSPCs with or without knockdown/deletion of CNOT3 would be valuable it is likely not easy; but one would expect an engraftment defect. The in vivo AML data is intriguing, however, one has to note that CNOT3 was again deleted prior to engraftment. One cannot distinguish homing from proliferative defects. While a reasonable approach, this should be discussed.

2) The authors show that CNOT3 affects c-MYC translation but not its expression. Next, they analyze codon frequency in transcripts that are differentially regulated by CNOT3 and suggest that CNOT3 may regulate a particular subset of transcripts that are more proliferative. They subsequently suggest translational regulation. If CNOT3 indeed regulates expression of transcripts with different codon composition, how does it do this? This should be analyzed and explained – is it via regulation of transcript stability?

Transcriptome, proteome data should be integrated. What is the overlap?

Minor:

1) Is it possible that the "differentiation" effect seen in AMLs is due to death of the more proliferative immature cells, skewing post treatment cells towards less proliferative, more mature cells?

2) Figure 1 U, V – legends are missing within figure

3) CFU assays for normal CD34+ should show serial replating; results should be discussed

4) Extended data figure 1W: why do the authors double stain for human CD45? Do the antibodies target the same or different epitopes?

5) Extended data figure 2A: LMPP and CLP are not shown.

Reviewer #3 (Remarks to the Author):

In the study by Ghashghaei et al., the authors meticulously investigate the role of CNOT3, a subunit of the CCR4-NOT complex, in the progression and development of acute myeloid leukemia (AML). Utilizing a combination of CRISPR/Cas screens, protein analysis, and various other molecular techniques, the authors discover the significant involvement of CNOT3 in leukemogenesis, with a particular focus on its modulation of translational efficacy.

The principal findings are as follows:

The CCR4-NOT complex, especially the CNOT3 subunit, is paramount in AML. CNOT3 exhibits increased expression in both AML cell lines and patient samples compared to healthy cells. There is a distinct elevation of CNOT3 amongst the CCR4-NOT subunits.

Several members of the CCR4-NOT complex, notably CNOT1, CNOT2, CNOT3, and CNOT10, are critical for the survival of leukemia cells in both human and mouse models. Intriguingly, the enzymatic members of this complex aren't as pivotal, hinting at potential redundancy or independence from its deadenylase activity.

Through a genome-wide CRISPR screen, CNOT3 was identified as crucial for AML cell survival.

Alterations in CNOT3 levels directly affect AML cell growth, differentiation, apoptosis, and morphology. Depletion results in the inhibition of cell growth and increased apoptosis, whereas its overexpression promotes leukemia cell proliferation.

The significance of CNOT3 in leukemogenesis is further reinforced through investigations using mouse models and primary AML cells.

CNOT3 is instrumental in preserving the undifferentiated states of human hematopoietic stem/progenitor cells (HSPCs).

The NOT box domain in CNOT3 is indispensable for enhancing leukemia cell growth.

Translational regulation of c-MYC expression is governed by CNOT3.

CNOT3-mediated gene expression is heavily influenced by codon usage and GC content.

CNOT3 seems involved in translation, as evidenced by its associations with ribosomal proteins and elongation factors.

A negative correlation exists between high CNOT3 expression and favorable prognosis in AML patients.

The principal weakness of this otherwise very well-performed study is that the mechanism by which CNOT3 affects leukemogenesis is not identified. CNOT3 is required for cell proliferation, but whether it is a causative factor rather than a strongly contributing one is unclear from this work. The overexpression result is interesting, but the authors made little use of it. The regulation of MYC translation by CNOT3 is not shown to be direct or indirect. And if it is direct it is not shown to be due to codon optimality rather than an alternative mechanism.

I should note, however, that the study is particularly timely as it connects a clinical implication for CNOT3 dysregulation with very recent work elucidating how CNOT3 targets stalled ribosomes and monitors for codon optimality at the translational level.

One issue that generally affects many studies, including the present one, that focus on the contributions of the CCR4-NOT subunits is it is often forgotten or missed entirely that they are pretty interdependent. The NOT box of CNOT3 is critical to the stability of the entire CCR4-NOT complex. This has been shown in several cellular and biochemical studies, and I would urge the authors to consult these and re-evaluate the discussion of their results in light of how the CCR4-NOT functions as a complex in which the deletion or manipulation of one of the subunits, or part of it, may adversely affect the entire complex. In general, there is over-reliance on the findings from the studies of the yeast system (presumably because codon optimality was demonstrated to be associated with the CCR4-NOT in yeast).

The following are mere suggestions for the authors that they may find helpful in improving their manuscript. These are not requirements for further experiments for resubmission.

As I pointed out above, the study provides evidence of the association of CNOT3 with translation machinery and changes in gene expression upon CNOT3 knockdown. However, the precise molecular mechanisms underlying these observations are not fully elucidated. Further experiments, such as ribosome profiling to assess translation efficiency or assays to determine how CNOT3 affects ribosome stalling, would provide more mechanistic insights.

The study primarily relies on the depletion of CNOT3 to draw conclusions. It would strengthen the case

to perform rescue experiments by reintroducing CNOT3 in cells with CNOT3 knockdown to confirm that the observed phenotypes are due to CNOT3 depletion and not off-target effects. Employing CRISPR/Cas9 gene editing for complete knockouts of CNOT3 will likely give more definitive results.

While the study identifies proteins that interact with CNOT3, it does not explore the functional significance of these interactions. Further experiments could investigate the role of specific interacting proteins in mediating CNOT3's effects on translation and gene expression.

There are correlations between CNOT3 expression and AML patient prognosis; it would be strengthened by functional experiments using patient-derived samples to link CNOT3 levels with disease outcomes directly. This would provide more clinical relevance to the findings.

The study relies on a limited number of cell lines and patient samples. To strengthen the conclusions, experiments should be replicated in a more extensive and diverse set of cell lines and patient-derived samples to account for biological variability.

While the study uses mouse models to examine the effects of CNOT3 depletion on leukemia development, additional *in vivo* experiments with different AML subtypes and patient-derived xenograft models would be very interesting.

The focus is on the short-term effects of CNOT3 depletion. Investigating the long-term consequences of CNOT3 loss on leukemia development and progression would provide a more comprehensive understanding of its role.

Mass spectrometry data may have false positives and negatives. Validation of specific protein interactions using orthogonal methods like co-immunoprecipitation or proximity ligation assays would enhance the reliability of the proteomic findings.

Below, I make recommendations for aspects of the study that the authors should address in resubmission.

In Figures 1E, 2E, 4D, EFig 4C, and D, the CNOT3 levels in KD-33 panels do not look too different from controls, yet the effects are almost as strong as in the case of KD-37 panel where KD looks far more efficient. The authors should consider quantifying the panel as a ratio to the control and perform a dilution series to show the dose-dependent effect. If this is not possible due to sample availability, authors should at least acknowledge and discuss the apparent difference. The western (particularly in the actin lanes) is questionable, which makes comparisons of the AML samples to the control difficult, given the need to normalize the samples to loading controls. Additionally, this experiment does not appear to have replicates, weakening its reliability substantially.

Lines 221-223: "It is recently reported that CNOT3 can directly interact with stalled ribosomes via its N-term domain while binding to several subunits of the CCR4-NOT complex including CNOT1, CNOT2, CNOT4, and CNOT10/11 via the NOT domain (Absmeier et al., 2022, bioRxiv)."

CNOT3 does not use the NOT domain to bind the CNOT4, CNOT10, or CNOT11. As noted above, I point the authors to the recent work on mammalian CCR4-NOT and recent reviews to equip themselves with facts regarding subunit interactions. The cited preprint (now published) does not make these claims.

Lines 523-525: "It is possible that a mechanism similar to what is described in yeast [Buschauer et al 2020] can also be employed in mammalian cells to couple translational control and mRNA stability." The work mentioned above has addressed this question.

Reference 23, a review from 2016, is now somewhat outdated. Authors are urged to cite more recent work wherever possible.

Figure 1: Expression of CNOT3 is bimodal in the normal CD34+ sample. Is the median fluorescent intensity plotted in F1D for the CNOT3+ population or the total population? Does the knockdown of CNOT3 affect the stability of the essential proteins CNOT1 and CNOT2?

Tumor engraftment analysis and Kaplan-Meier curves for control and MOLM13 cells with CNOT3 overexpression would be helpful if the authors have these.

Figures 1U and 1V should include a legend either in the figure or in the figure caption describing the difference between the red and blue curves.

Line 171: The term 'driving' is not justified. Figure 1 shows that CNOT3 is essential for leukemogenesis but does not demonstrate that CNOT3 is driving leukemogenesis.

Why are the shRNAs called KD-33 and KD-37 is this an internal lab code? Is it relevant to our interpretation of the data?

Figure 2: Why does the control in F2B increase in cell number to a greater extent than the control in F2G in the same timeframe?

Extended data Figure 2 should include a plot like EDF2C for F2I depicting the gating strategy for CD11b, CD14, and CD13 for the empty vector control and the CNOT3 overexpression condition.

Figure 4: The authors should show that CNOT3 directly rather than indirectly regulates MYC translation. They should demonstrate that CNOT3 associates with MYC mRNA during translation.

Is there a correlation between CNOT3 protein and MYC protein in AML patient samples?

Does overexpression of CNOT3 in MOLM13 cells or primary HSPCs increase the MYC protein expression level without an impact on MYC mRNA levels?

Figure 5: Can the authors demonstrate that the codon optimality of MYC mRNA influences the regulation of translation by CNOT3?

Figure 6: It would strengthen the case substantially to compare protein binding partners of CNOT3 mutants lacking either the N-term coiled-coil. The deletion of the C-terminal NOT box would destabilize the entire CCR4-NOT. Do the authors anticipate that CNOT3 lacking the NOT box is still able to associate with ribosomes but unable to modulate translation?

Figure 7: Their model implies that mRNA is under 3 prime to 5 prime decay at the same time as undergoing increased translation, but this is not supported by their data.

Syntax and readability issues:

Line 341: 'for examples'

Line 373: 'among the translationally downregulate 'transcript'

There is a mixture of cMyc and c-Myc across texts and figures.

The word "diseases" in line 91 should not be plural

The sentence in lines 495-497 is not grammatically correct and instead should read, "a pattern emerged showing genes that/which exhibited low G/C and high A/T content and enriched in some rare codons (i.e., lysine) to be preferentially promoted by CNOT3."

Eugene Valkov, National Cancer Institute, U.S.A.

Reviewer #1 (Remarks to the Author):

Review of Manuscript NCOMMS-23-41711-T by Dr. Vu and Co-Authors Titled "Translation Efficiency Driven by the CNOT3 Subunit of the CCR4-NOT Complex Promotes Leukemogenesis"

In this study, Dr. Vu and their co-authors have presented compelling evidence regarding the pivotal role of CNOT3 in enhancing the translation of c-MYC, thereby promoting leukemogenesis. The authors skillfully demonstrated that the expression of the CNOT3 subunit within the CCR4-NOT complex is significantly upregulated in Acute Myeloid Leukemia (AML). They convincingly showed that CNOT3 actively fosters the survival and proliferation of leukemia cells, as effectively depicted in Figure 1. Furthermore, the authors established that CNOT3 is indispensable for leukemogenesis and the maintenance of undifferentiated states within human hematopoietic stem/progenitor cells (HSPCs), as visually presented in Figure 2. Notably, they identified the NOT box domain as a crucial component for CNOT3 function in leukemia cells, as highlighted in Figure 3. Additionally, the authors provided valuable insights into how CNOT3 regulates c-MYC expression at the translational level, as exemplified in Figure 4. They also conducted a comprehensive global assessment of CNOT3 mRNA targets, as demonstrated in Figure 5.

The manuscript's exploration of the unique and intriguing translation regulation of c-MYC by the major deadenylase CCR4-NOT is of significant interest to the readership of Nature Communications. However, further mechanistic insights are warranted. I recommend the publication of this study in Nature Communications, contingent upon the incorporation of additional validation of CNOT3's function in c-MYC translation. I would like to raise a couple of important points for consideration:

We sincerely thank the reviewer for recognizing the rigorousness and significance of our work.

While the study has made significant strides in uncovering CNOT3's role in c-MYC translation, its mechanistic insights remain elusive. Given the essential role of the NOT box domain in the CNOT3 function (Figure 3D), it is imperative to conduct polysome analysis of c-MYC in the context of NOT box-truncated CNOT3, as illustrated in Figure 4H. Furthermore, investigating the association of the NOT box-truncated CNOT3 with the translation machinery, as outlined in Figure 6, is also crucial to provide additional valuable insights.

We agree with the reviewer that further exploration into the contribution of the NOT box domain in CNOT3 function provides additional insights into the mechanisms of action of CNOT3 in leukemia cells.

As suggested by the reviewer, we performed immunoblot for c-MYC and polysome profiling and analysis of c-MYC mRNA in the context of NOT box truncated CNOT3. We observed that overexpression of CNOT3 did not drive more c-MYC expression (data shown below). Neither

overexpression of the CNOT3 nor the NOT box truncated CNOT3 impacted *c-MYC* mRNA and protein expression or association of *c-MYC* transcript with the polysome (vs. monosome) fractions. *c-MYC* is considered one of the most potent oncogenes and its expression is tightly regulated in cancer cells. Therefore, it is possible that further induction of CNOT3 activity is restrained by other mechanisms to maintain *c-MYC* optimal expression level.

To further address the association of the NOT box-truncated CNOT3 with the translation machinery, we used the CNOT3 FL and NOT box truncated overexpressing cell lines to perform PLA using Flag antibody paired with antibodies raised against puromycin, EEF1D and EEF1E to specifically examine CNOT3 association between the elongating peptides as well as EEF1D and EEF1E. Despite some non-specific signals with Flag antibody in the empty vector (EV) control cells, we observed that overexpression of CNOT3-FL significantly enhanced the PLA signals. On the other hand, PLA signals detected in cells carrying the NOT box truncated CNOT3 were similar to those in EV control. The results indicate that deletion of the NOT box domain compromised

association of CNOT3 with the translation machinery. The data shown below is also now included in the manuscript as extended data figure 6D-I

Flag(CNOT3)-Puro-PLA

Flag(CNOT3)-EEF1E-PLA

Flag(CNOT3)-EEF1D-PLA

Although the authors have convincingly demonstrated the essential role of the NOT box domain in CNOT3 function in AML (Figure 3D), **it is advisable** to explore whether the N-terminal domain of CNOT3 also plays a crucial role in its function and its impact on c-MYC translation in AML. I strongly encourage the authors to expand their investigation of c-MYC translation (Figure 4) and the global assessment of CNOT3 (Figure 5). This additional analysis would contribute to a more comprehensive understanding of the mechanisms underlying CNOT3-mediated translation regulation.

We truly appreciate the advices from the reviewer and have taken into consideration all suggestions for our future studies to follow up on the findings presented in the current manuscript.

Reviewer #2 (Remarks to the Author):

In this manuscript Ghashghaei et al. dissect the role of CNOT3 and the CCR4-NOT complex in hematopoiesis and leukemogenesis. They identify CNOT3 as overexpressed in 3 AML datasets which peaks their interest in this protein and whether it may specifically contribute to leukemia survival. They test several AML cell lines, all of which express CNOT3 protein with higher expression in some than in normal human cord blood hematopoietic stem and progenitor CD34+ cells. They show that knockdown of CNOT3 via shRNA and deletion via CRISPR results in reduced AML growth using various AML models. Knockdown in normal CD34+ results in lack of differentiation and reduced colony formation. Overexpression has the opposite effects. Next they perform CRISPR tiling to identify the domains of CNOT3 responsible for this effect. They identify N- and C-terminal domains of interest, but focus subsequently on the C-terminal NOT box domain. They confirm functionally, that the NOT box domain is essential for CNOT3's function.

They subsequently perform transcription and proteomic profiling of control and CNOT3 knockdown leukemia cells and identify c-MYC as CNOT target. Various techniques serve to dissect the mechanism of this regulation, suggesting the c-MYC translation is affected by CNOT3 KD.

Given that CNOT3 participates in the CCR4-NOT complex mediated translational control, they next analyzed transcripts that were differentially regulated for nucleotide usage and codon composition. They suggest that CNOT3 regulates expression of genes based on bias at the third "wobble" codon that correspond to the cell's proliferative state. Via Ribo-STAMP they also analyze the translational efficiency in wildtype and CNOT3 knockdown cells suggesting that CNOT3 may enhance translation of highly translated mRNAs. Indeed, via CNOT3 Co-IP followed by mass spec and validation via the Puro-PLA assay, they validate association of CNOT3 with the translational machinery.

Overall, this is a highly informative and very well executed study. The authors very thoroughly dissect the role of CNOT3 in leukemogenesis and carefully analyze its mechanistic role in mammalian cells. The data is very well presented.

We thank the reviewer for the positive evaluation and the very thoughtful summary of our work.

Two conceptual questions arise:

1) The authors suggest that CNOT3's role is unique to AML and that its targeting could represent a therapeutic opportunity. They do not determine how CNOT3 and CCR4-NOT machinery is upregulated in AML. This should be discussed.

We thank the reviewer for bringing up the relevant points. We agree that it would be of interest to understand how CNOT3 (and possibly other subunits of the CCR4-NOT complex) is upregulated in AML. To address this question, we first investigated whether there is any correlation between CNOT3 expression with genetic alternations linked to AML pathogenesis. We

hypothesized that frequently recurring genetic alterations may drive *CNOT3* upregulation. We examined the expression of *CNOT3* across different subtypes of AML in TCGA and beatAML datasets. Except for complex karyotype AML subgroup in beatAML database showing a statistically significant higher *CNOT3* level, we did not observe any particular association between *CNOT3* expression and specific gene mutations (data shown below), suggesting that elevated *CNOT3* expression is a general phenomenon across AML subtypes.

While it remains an important question in the field, it has been postulated that expression of some RBPs can be influenced by transcription factors (TFs), which in turn orchestrate upregulation of the post-transcriptional circuitry to enforce expression of the oncogenic gene programs. To further look into the possible association of *CNOT3* expression with TFs, we performed TF binding analysis and found 30 TFs predicted to bind to *CNOT3* promoter and are positively associated with *CNOT3* expression on the BeatAML dataset. Among them are TFs which are known to promote AML such as *HOX9A*, *ETV6* and the TF that showed the strongest correlations across the most AML subtypes was *NFIC* (data shown below). Thus, it is possible that elevated activities of these TFs upregulate *CNOT3* expression in AML. Further experimental study is required to conclusively test the hypothesis and demonstrate the functional links and it is beyond the scope of our current manuscript.

BeatAML dataset

The data on CD34+ cells is somewhat limited, restricted to umbilical cord blood (instead of adult CD34+) and liquid culture and CFU assays without serial replating. The latter should be performed to assess not only progenitors but more immature cells. While engraftment of HSPCs with or without knockdown/deletion of CNOT3 would be valuable it is likely not easy; but one would expect an engraftment defect. The in vivo AML data is intriguing; however, one has to note that

CNOT3 was again deleted prior to engraftment. One cannot distinguish homing from proliferative defects. While a reasonable approach, this should be discussed.

We thank the reviewer for being very considerate and recognizing the technical challenges involved in an *in vivo* assay.

We repeated the colony forming assay and performed serial replating to further assess the consequences of CNOT3 depletion. We reproducibly demonstrated that efficient CNOT3 knockdown impaired colony forming ability of HSPCs at the first plating. However, upon examining the efficacy of CNOT3 knockdown of cells after 10 days of the first round of plating, we found that CNOT3 was no longer effectively depleted (data shown below), thus resulting in no difference in the ability to form colonies between cells transduced with control vs. KD-33 and KD-37 shRNAs. The technical caveat is likely due to either incomplete selection by puromycin of shRNA-transduced cells or silencing of shRNAs over the period of 10 days of the colony forming assay. Therefore, it is not possible for us to accurately assess the impact of CNOT3 depletion on replating of human HSPCs.

To address the issue, we took advantages of the genetic murine model of conditional *Cnot3* knockout (KO) available in our lab (unpublished data-manuscript in preparation). Deletion of *Cnot3* in the blood system is mediated by Mx-1 Cre. Using this model, we observed that deletion of *Cnot3* reduced the ability of bone marrow cells to replate (confidential data shown below).

CB-CD34+ cells harvested from 1st colony plating

Mouse Bone Marrow cells isolated from WT vs. *Cnot3* KO mice

We have also added the sentence “*While the in vitro and in vivo phenotypes are highly agreeable, it remains to be determined whether loss of CNOT3 can lead to any homing defects in both normal and leukemia cells*” in the discussion to note the potential impacts of CNOT3 depletion on the homing ability of leukemia cells and HSPCs.

2) The authors show that CNOT3 affects c-MYC translation but not its expression. Next, they analyze codon frequency in transcripts that are differentially regulated by CNOT3 and suggest that CNOT3 may regulate a particular subset of transcripts that are more proliferative. They subsequently suggest translational regulation. If CNOT3 indeed regulates expression of transcripts with different codon composition, how does it do this? This should be analyzed and explained – is it via regulation of transcript stability?

We thank the reviewer to point out these points.

While it is appealing to make a direct connection between codon composition, codon optimal and mRNA decay, we would like to exercise some cautions toward making that claim. This is due to the fact that the codon optimality must be determined for each and every cell type based on the composition and availability of tRNAs as well as conditions of culture such as supplies of amino acids. A transcriptomic wide profiling of RNA half-life and global profiling of tRNAs in leukemia cells are beyond the scope of our current manuscript.

On the other hand, we looked into the AT vs. GC composition of the top 10 downregulated and upregulated genes upon CNOT3 depletion and found that there are several genes exhibiting more enriched AT vs. GC representation. These includes upregulated genes *BTG2* and *AXL* with high GC% and downregulated genes *SHLD2*, *KIF5B* and *PTGS3* with low GC% (data shown below). To test the hypothesis that the expression of some transcripts with high AT vs. GC percentage might be regulated via RNA stability, we picked these candidates genes and evaluated RNA stability upon CNOT3 knockdown. After actinomycin D treatment, we tracked mRNA abundance over the period of 4 hours (240 minutes). We observed that while half-life of *SHLD2*, *KIF5B* and *PTGS3* mRNAs significantly decreased upon CNOT3 knockdown, *BTG2* and *AXL* exhibited increased or no change in stability (data shown below). The results supported the notion that regulation of RNA stability contributes to CNOT3’s control of gene expression.

Transcriptome, proteome data should be integrated. What is the overlap?

We overlapped the downregulated and upregulated genes/proteins captured in our RNA-seq and mass spectrometry analysis of the transcriptome (FDR ≤ 0.05 ; foldchange (FC) < 0 or > 0) and the proteome (FDR ≤ 0.1 ; foldchange (FC) < 0 or > 0) upon CNOT3 depletion. Given that the coverage of the proteome is limited in comparison to the transcriptome, we observed $\sim 32\%$ and $\sim 46\%$ of proteins exhibited correlative changes with downregulated and upregulated mRNAs respectively.

We performed pathway analysis and found that these genes/proteins are similarly enriched in pathways identified previously to be shared between transcriptomic and proteomic analysis. The results further supported the idea that despite the imperfect correlation between mRNAs and proteins, there is coordination in control of mRNA and protein abundance to control gene expression.

Minor:

1) Is it possible that the “differentiation” effect seen in AMLs is due to death of the more proliferative immature cells, skewing post treatment cells towards less proliferative, more mature cells?

Differentiation is coupled with cell division and proliferation. Since it is technically very challenging to run flow analysis for both differentiation markers and apoptotic marker Annexin V, we are unable to draw a definite conclusion. However, a block in differentiation is the hallmark of leukemia cells. Therefore, forcing differentiation in leukemia cells subsequently leads to cell death so the two processes likely occur simultaneously in leukemia cells. On the other hand, we did not observe increase in apoptosis in HSPCs undergoing differentiation upon CNOT3 knockdown. This suggests that CNOT3 depletion can promote differentiation without inducing death at least in normal stem/progenitor cells.

2) Figure 1 U, V – legends are missing within figure

We have added the missing legends to the figure.

3) CFU assays for normal CD34+ should show serial replating; results should be discussed

Please see response to point (1) in the major comment section.

4) Extended data figure 1W: why do the authors double stain for human CD45? Do the antibodies target the same or different epitopes?

The two antibodies against human CD45 target different epitopes. By using the two antibodies, we increased specificity of the staining for engrafted human cells in recipient mice. The approach allowed us to detect a clear and well separated population of human leukemia cells as shown in supplemental figure 1W.

5) Extended data figure 2A: LMPP and CLP are not shown.

The results shown in extended data figure 2A were taken from bloodspot analysis of several datasets GSE17054; GSE19599; GSE11864 and E-MEXP-1242. LMPP or CLP data was not available in the analysis. Therefore, we were not able to include LMPP and CLP in the comparison.

Reviewer #3 (Remarks to the Author):

In the study by Ghashghaei et al., the authors meticulously investigate the role of CNOT3, a subunit of the CCR4-NOT complex, in the progression and development of acute myeloid leukemia (AML). Utilizing a combination of CRISPR/Cas screens, protein analysis, and various other molecular techniques, the authors discover the significant involvement of CNOT3 in leukemogenesis, with a particular focus on its modulation of translational efficacy.

The principal findings are as follows:

The CCR4-NOT complex, especially the CNOT3 subunit, is paramount in AML. CNOT3 exhibits increased expression in both AML cell lines and patient samples compared to healthy cells. There is a distinct elevation of CNOT3 amongst the CCR4-NOT subunits.

Several members of the CCR4-NOT complex, notably CNOT1, CNOT2, CNOT3, and CNOT10, are critical for the survival of leukemia cells in both human and mouse models. Intriguingly, the enzymatic members of this complex aren't as pivotal, hinting at potential redundancy or independence from its deadenylase activity.

Through a genome-wide CRISPR screen, CNOT3 was identified as crucial for AML cell survival. Alterations in CNOT3 levels directly affect AML cell growth, differentiation, apoptosis, and morphology. Depletion results in the inhibition of cell growth and increased apoptosis, whereas its overexpression promotes leukemia cell proliferation.

The significance of CNOT3 in leukemogenesis is further reinforced through investigations using mouse models and primary AML cells.

CNOT3 is instrumental in preserving the undifferentiated states of human hematopoietic stem/progenitor cells (HSPCs).

The NOT box domain in CNOT3 is indispensable for enhancing leukemia cell growth. Translational regulation of c-MYC expression is governed by CNOT3.

CNOT3-mediated gene expression is heavily influenced by codon usage and GC content.

CNOT3 seems involved in translation, as evidenced by its associations with ribosomal proteins and elongation factors.

A negative correlation exists between high CNOT3 expression and favorable prognosis in AML patients.

We thank the reviewer for a very positive and thorough summary of the main findings of our work.

The principal weakness of this otherwise very well-performed study is that the mechanism by which CNOT3 affects leukemogenesis is not identified. CNOT3 is required for cell proliferation, but whether it is a causative factor rather than a strongly contributing one is unclear from this work. The overexpression result is interesting, but the authors made little use of it.

We appreciate the reviewer for noting that “the result is interesting”. We observed that overexpression of CNOT3 can promote proliferation in both leukemia cells and normal hematopoietic stem/progenitor cells. The data strongly suggested that elevated expression of CNOT3 can at least in part contribute to a growth advantage. Determining whether CNOT3 can cooperate with other oncogenic events to initiate and drive more aggressive diseases is of interest and we agree that it will be an excellent direction for a follow up study.

The regulation of MYC translation by CNOT3 is not shown to be direct or indirect. And if it is direct, it is not shown to be due to codon optimality rather than an alternative mechanism.

We agree with the review that it is beneficial to determine whether the regulation is direct or indirect. To this end, we performed RNA-immunoprecipitation followed by qPCR to determine whether CNOT3 binds to *c-MYC* mRNA. We found that there is a trend in increased enrichment with CNOT3 pulldown vs. IgG control (data shown below) but the interaction is likely weak. In our opinion, this is fairly reasonable as CNOT3 by itself is not an RNA binding protein (RBP). It is known that CNOT3 association with mRNAs can be mediated by other RBPs and its interaction with the CCR4-NOT complex.

Regarding codon optimality, while it is appealing to make a direct connection between codon composition, codon optimal and mRNA decay, we would like to exercise some cautions toward making that claim. This is due to the fact that the codon optimality must be determined for each and every cell type based on the composition and availability of tRNAs as well as conditions of cell culture in particular supplies of amino acids. Further inquiries into this area of research is well justified but is beyond the scope of our current manuscript.

I should note, however, that the study is particularly timely as it connects a clinical implication for CNOT3 dysregulation with very recent work elucidating how CNOT3 targets stalled ribosomes and monitors for codon optimality at the translational level.

We thank the reviewer for the compliment and to recognize our work as “timely”.

One issue that generally affects many studies, including the present one, that focus on the contributions of the CCR4-NOT subunits is it is often forgotten or missed entirely that they are pretty interdependent. The NOT box of CNOT3 is critical to the stability of the entire CCR4-NOT complex. This has been shown in several cellular and biochemical studies, and I would urge the authors to consult these and re-evaluate the discussion of their results in light of how the CCR4-NOT functions as a complex in which the deletion or manipulation of one of the subunits, or part of it, may adversely affect the entire complex. In general, there is over-reliance on the findings from the studies of the yeast system (presumably because codon optimality was demonstrated to be associated with the CCR4-NOT in yeast).

We agree with the reviewer’s remark on the importance of evaluating proteins’ function within the context of their functional complex(es). We performed additional immunoblots to determine whether CNOT3 depletion affects stability and abundance of CNOT1 protein. We observed a slight change in CNOT1 expression in KD-37 samples. However, reduction of CNOT3 by ~40% using shRNA- KD-33, which significantly reduced c-MYC expression and impaired cell growth, did not significantly reduce CNOT1 expression (data shown below). While the data does not capture the stability of the whole complex, it supports the notion that at least at the earlier time point of our assessments, given the stable expression of the major scaffolding subunit CNOT1, the observed phenotypes upon CNOT3 depletion are the results of direct inhibition of CNOT3 function in leukemia cells.

In addition, we have revised the manuscript to include additional discussion to take in consideration the interaction of CNOT3 and its role within the CCR4-NOT complex: “CNOT3 is a conserved subunit of the CCR4-NOT complex where it makes stable interactions with the scaffolding subunit CNOT1 and subunit CNOT2^{27,28}. It has been demonstrated that knockdown of

CNOT3 results reduced expression of other subunits of the complex including CNOT1 and CNOT2⁷². In our study, we observed minimal impact on CNOT1 protein abundance at least at the early time points when CNOT3 was depleted in leukemia cells and HSPCs (data not shown). While CNOT3 depletion might not have strong impact on total protein abundance of other subunits in our system, it is possible that it might still impair assembly and stability of the complex, hence adversely affect functions of the entire complex."

The following are mere suggestions for the authors that they may find helpful in improving their manuscript. **These are not requirements for further experiments for resubmission.**

We sincerely thank the reviewer for being very considerate in giving suggestions and advices for us to improve the manuscript. Given the timeframe provided for the revision, we have addressed several points with additional experimental results while including additional discussion to provide better contexts and potential areas for future investigations.

As I pointed out above, the study provides evidence of the association of CNOT3 with translation machinery and changes in gene expression upon CNOT3 knockdown. However, the precise molecular mechanisms underlying these observations are not fully elucidated. Further experiments, such as ribosome profiling to assess translation efficiency or assays to determine how CNOT3 affects ribosome stalling, would provide more mechanistic insights.

We thank the reviewer for the suggestion. Even though ribosome profiling can provide better resolution for assessments of ribosome stalling, the requirement of large amount of input RNA (~at least 100 million leukemia cells) to perform a robust ribosome profiling poses a significant technical challenge for us to implement in the study. Therefore, we chose to use the two complementary approaches i.e., polysome profiling followed by qPCR and RiboStamp, which is more amenable for downstream functional evaluation. While we will not be able to conclude on the impact of CNOT3 on ribosome stalling, the results obtained by polysome profiling and RiboStamp sufficiently supported CNOT3's function in translation control in leukemia.

The study primarily relies on the depletion of CNOT3 to draw conclusions. It would strengthen the case to perform rescue experiments by reintroducing CNOT3 in cells with CNOT3 knockdown to confirm that the observed phenotypes are due to CNOT3 depletion and not off-target effects. Employing CRISPR/Cas9 gene editing for complete knockouts of CNOT3 will likely give more definitive results.

We thank the reviewer to point this out.

In fact, we performed the rescue experiment by overexpressing CNOT3 in CNOT3-depleted leukemia cells. We demonstrated that CNOT3 overexpression at least partially rescued proliferative defects when CNOT3 was knocked down. The data was included in Extended data figure 1T in the submitted manuscript.

We also performed CRISPR/Cas9 mediated knockout of CNOT3 to validate the observed phenotypes using shRNA-mediated knockdown of CNOT3. We used two independent Cas9 leukemia cell lines i.e., MOLM13-Cas9 and MV4-11- Cas9 and demonstrated that CRISPR/Cas9 knockout of CNOT3 also led to inhibition of cell growth, induction of differentiation and apoptosis. The results were included in Extended data figure 1J-Q in the submitted manuscript.

While the study identifies proteins that interact with CNOT3, it does not explore the functional significance of these interactions. Further experiments could investigate the role of specific interacting proteins in mediating CNOT3's effects on translation and gene expression.

We thank the reviewer for the comment.

To address these questions, we performed shRNA-mediated knockdown of the two elongation factors *EEF1G* and *EEF1E1* and examined the impacts of *EEF1G* and *EEF1E1* depletion on cell growth and viability as well as c-MYC expression. We found that efficient knockdown of *EEF1G* and *EEF1E1* expression resulted in significant inhibition of cell proliferation and increased apoptosis and a marked reduction in c-MYC protein level. These data support the involvement of these translation factors in promoting leukemia cell survival and expression of c-MYC. The data shown below is also included in the revised manuscript as figure 6L-P.

There are correlations between CNOT3 expression and AML patient prognosis; it would be strengthened by functional experiments using patient-derived samples to link CNOT3 levels with disease outcomes directly. This would provide more clinical relevance to the findings.

We agree with the review regarding the importance of evaluating CNOT3 function using patient-derived samples. Indeed, we performed a number of experiments using primary AML patient samples. These includes immunoblots and intracellular staining to examine CNOT3 protein expression in primary AML patients (data shown in figure 1B-D); shRNA-mediated knockdown of CNOT3 in primary samples for *in vitro* liquid culture and colony forming assay (data shown in figure 1N-P) and depletion of CNOT3 in 3 patient derived xenografts (PDXs) for *in vivo* assessment of leukemogenesis (data shown in figure 1Q-T and extended data figure 1W-X).

The study relies on a limited number of cell lines and patient samples. To strengthen the conclusions, experiments should be replicated in a more extensive and diverse set of cell lines and patient-derived samples to account for biological variability.

We have extensively evaluated consequences of CNOT3 depletion in a panel of 7 AML cell lines with diverse genetic backgrounds including MOLM13, MV4-11, OCI-AML3, NB4, HL-60, NOMO-1 and THP-1 (data shown in figure 1E-I and extended data figure 1E-Q). We performed functional assessments using a total of 6 primary patient samples and 3 independent PDXs. While it is desirable to include more samples, our work covered a significant range of cell lines as well as primary samples and strongly comparable with other studies in the field.

While the study uses mouse models to examine the effects of CNOT3 depletion on leukemia development, additional *in vivo* experiments with different AML subtypes and patient-derived xenograft models would be very interesting. The focus is on the short-term effects of CNOT3 depletion. Investigating the long-term consequences of CNOT3 loss on leukemia development and progression would provide a more comprehensive understanding of its role.

In this study, we used NSG and NRG-3GS mice, which are the mouse strains used to host human cells for *in vivo* studies. We would like to point out that it is distinctively different from genetic murine models. In fact, in agreement with the review's comment, we used patient derived xenografts (PDXs) to evaluate the consequences of CNOT3 loss of function in leukemia development *in vivo* (the data shown in figure 1Q-T). *In vivo* engraftment of leukemia cells also allowed us to track leukemia development over the long period of time – from at least 8 weeks (data shown in figure 1R-T) to up to 120 days (data shown in figure 1M).

Mass spectrometry data may have false positives and negatives. Validation of specific protein interactions using orthogonal methods like co-immunoprecipitation or proximity ligation assays would enhance the reliability of the proteomic findings.

We thank the review for the comment. To address this point, we performed PLA using antibodies against CNOT3 and EEF1E and EEF1D. The selection of the elongation factors to perform the assay is based on the availability of antibodies raised in rabbit for EEF1E and EEF1D to pair with CNOT3 mouse antibody. We observed strong PLA signals indicating association of CNOT3 with both EEF1E and EEF1D. The data shown below is also included in the revised manuscript as figure 6H-K.

In addition, we also used the CNOT3 FL and NOT box truncated overexpressing cell lines to perform PLA using Flag antibody paired with antibodies raised against EEF1D and EEF1E to examine CNOT3 association between the elongating peptides as well as EEF1D and EEF1E (data shown in the later section in response to the reviewer's comment on Figure 6). The data further supported PLA results of endogenous proteins and indicated that the NOT box domain of CNOT3 is essential for the association of CNOT3 with these factors.

Below, I make recommendations for aspects of the study that the authors should address in resubmission.

In Figures 1E, 2E, 4D, EFig 4C, and D, the CNOT3 levels in KD-33 panels do not look too different from controls, yet the effects are almost as strong as in the case of KD-37 panel where KD looks far more efficient. The authors should consider quantifying the panel as a ratio to the control and perform a dilution series to show the dose-dependent effect. If this is not possible due to sample availability, authors should at least acknowledge and discuss the apparent difference. The western (particularly in the actin lanes) is questionable, which makes comparisons of the AML samples to the control difficult, given the need to normalize the samples to loading controls. Additionally, this experiment does not appear to have replicates, weakening its reliability substantially.

We thank the review to bring up this point and allow us to provide multiple replicates to support the reproducibility of our work.

Replicates for Figure 1E and 4D

Replicates for Figure 2A

Replicates for Figure 3C

Lines 221-223: "It is recently reported that CNOT3 can directly interact with stalled ribosomes via its N-term domain while binding to several subunits of the CCR4-NOT complex including CNOT1, CNOT2, CNOT4, and CNOT10/11 via the NOT domain (Absmeier et al., 2022, bioRxiv)." CNOT3 does not use the NOT domain to bind the CNOT4, CNOT10, or CNOT11. As noted above, I point

the authors to the recent work on mammalian CCR4-NOT and recent reviews to equip themselves with facts regarding subunit interactions. The cited preprint (now published) does not make these claims.

We thank the reviewer to point this out. We have updated the citation to reflect the publication of the preprint. We also double checked the reported results where it was described that “First, the CNOT3 NTR crosslinked to the CNOT10/11 module and the CNOT4 NTR. This agrees with these proteins being in close proximity on the 80S ribosome and therefore also to each other (Fig. 4). The CNOT3 NTR also crosslinked to CNOT2, CNOT1 and the exonuclease subunit CNOT6. The CTR of CNOT3 crosslinked to a number of subunits including CNOT2 and CNOT1 (Extended Data Fig. 6, c), in agreement with a previously reported crystal structure of the NOT module. The CNOT3 CTR also crosslinked to CNOT4 and CNOT9.” (Absmeier et al., 2023).

We therefore revised our text to accurately capture the interaction at both the N-term and C-term domains of CNOT3 *“It is recently reported that CNOT3 can directly interact with stalled ribosomes via its N-term domain while binding to several subunits of the CCR4-NOT complex including CNOT1, CNOT2, CNOT4, CNOT6 and CNOT10/11 via both the N-term domain and the C-term domain”*.

Lines 523-525: “It is possible that a mechanism similar to what is described in yeast [Buschauer et al 2020] can also be employed in mammalian cells to couple translational control and mRNA stability.” The work mentioned above has addressed this question.

We have revised the text to recognize the newly published study.

Reference 23, a review from 2016, is now somewhat outdated. Authors are urged to cite more recent work wherever possible.

We thank the reviewer for the comment. We have now included the more recent articles to better capture the current status of the field. These includes:

Collart, M.A., Audebert, L. & Bushell, M. Roles of the CCR4-Not complex in translation and dynamics of co-translation events. *Wiley Interdiscip Rev RNA*, e1827 (2023).

Raisch, T. & Valkov, E. Regulation of the multisubunit CCR4-NOT deadenylase in the initiation of mRNA degradation. *Curr Opin Struct Biol* 77, 102460 (2022).

Höpfler, M., et al. Mechanism of ribosome-associated mRNA degradation during tubulin autoregulation. *Mol Cell* 83, 2290-2302.e2213 (2023).

Figure 1: Expression of CNOT3 is bimodal in the normal CD34+ sample. Is the median fluorescent intensity plotted in F1D for the CNOT3+ population or the total population?

The median fluorescent intensity plotted in Figure 1D is for the total population.

Does the knockdown of CNOT3 affect the stability of the essential proteins CNOT1 and CNOT2?

As mentioned in the response to the comment above, we evaluated CNOT1 expression upon CNOT3 knockdown and saw a slight change in CNOT1 expression in KD-37 samples. Reduction of CNOT3 by ~40% using shRNA- KD-33 did not significantly reduce CNOT1 expression.

Tumor engraftment analysis and Kaplan-Meier curves for control and MOLM13 cells with CNOT3 overexpression would be helpful if the authors have these.

We agree that it is helpful to evaluate the impact in vivo. However, the timeframe of the revision does not allow for us to perform and follow this longer-term experiment. In addition, we believe that evaluate the impact of CNOT3 overexpression in the context of cooperation with other oncogenic events to initiate leukemogenesis will be more beneficial. Therefore, it is one of the future directions that we aim to pursue in our future studies.

Figures 1U and 1V should include a legend either in the figure or in the figure caption describing the difference between the red and blue curves.

We added the legends for Figure 1U and 1V.

Line 171: The term 'driving' is not justified. Figure 1 shows that CNOT3 is essential for leukemogenesis but does not demonstrate that CNOT3 is driving leukemogenesis.

We removed the word “driving” to better describe the observed phenotypes.

Why are the shRNAs called KD-33 and KD-37 is this an internal lab code? Is it relevant to our interpretation of the data?

The shRNAs' names do not have any significance to the interpretation of the data. KD-33 and KD-37 are our coded names for the two hairpins from the TRC clones: TRCN0000015133 and TRCN0000015137.

Figure 2: Why does the control in F2B increase in cell number to a greater extent than the control in F2G in the same timeframe?

In figure 2B and 2G, within the period of 7 days, the cells in the control conditions increased from 0.2 million cells at the time of plating (D0) to a range of 1.0-1.6 million cells. The number is within the expected doubling time of normal HSPCs. The slight difference between controls in Figure 2B and Figure 2G is likely due to the fact that they are two different lentiviral backbones. In figure 2B, the control is pLKO.1 carrying scramble shRNA and in figure 2G, the control is lentiviral pMNUD3 empty vector. In figure 2B, transduced cells were selected using puromycin and plated directly to culture after 3 days of selection. On the other hand, in figure 2G, transduced cells were sorted based on YFP positivity, prior to plating in media.

Extended data Figure 2 should include a plot like EDF2C for F2I depicting the gating strategy for CD11b, CD14, and CD13 for the empty vector control and the CNOT3 overexpression condition.

We added the representative flow plots with gating strategy (shown below) to Extended Data figure 2D.

Figure 4: The authors should show that CNOT3 directly rather than indirectly regulates MYC translation. They should demonstrate that CNOT3 associates with MYC mRNA during translation.

We performed RNA-immunoprecipitation followed by qPCR to determine whether CNOT3 binds to *c-MYC* mRNA. We found that there is a trend in increased enrichment with CNOT3 pulldown

vs. IgG control (data shown below) but the interaction is likely weak. In our opinion, this is fairly reasonable as CNOT3 by itself is not an RNA binding protein (RBP). It is known that CNOT3 association with mRNAs can be mediated by other RBPs and its interaction with the CCR4-NOT complex.

Is there a correlation between CNOT3 protein and MYC protein in AML patient samples?

We thank the reviewer for the comment.

We looked into *CNOT3* and *c-MYC* mRNA expression in the two TCGA and beatAML cohorts and found a significant positive correlation between *CNOT3* vs. *c-MYC* expression levels (data shown below).

Does overexpression of CNOT3 in MOLM13 cells or primary HSPCs increase the MYC protein expression level without an impact on MYC mRNA levels?

We performed immunoblot for c-MYC and polysome profiling and analysis of *c-MYC* in the context of NOT box truncated CNOT3. While CNOT3 depletion resulted in significant reduction in c-MYC (figure 4), we observed that additional exogenous expression of CNOT3 did not drive more c-

MYC expression and translation activity (data shown below). Neither overexpression of the CNOT3 nor the NOT box truncated CNOT3 impacted c-MYC mRNA and protein expression or association of c-MYC transcript with the polysome (vs. monosome) fractions. c-MYC is considered one of the most potent oncogenes and its expression is tightly regulated in cancer cells. Therefore, it is possible that further induction of CNOT3 activity is restrained by other mechanisms to maintain c-MYC optimal expression level.

Figure 5: Can the authors demonstrate that the codon optimality of MYC mRNA influences the regulation of translation by CNOT3?

While it is appealing to make a direct connection between codon composition, codon optimal and mRNA decay, we would like to exercise some cautions toward making that claim. This is due to the fact that the codon optimality must be determined for each and every cell type based on the composition and availability of tRNAs as well as conditions of cell culture in particular supplies of amino acids. Further inquiries into this area of research are well justified but is beyond the scope of our current manuscript.

Figure 6: It would strengthen the case substantially to compare protein binding partners of CNOT3 mutants lacking either the N-term coiled-coil. The deletion of the C-terminal NOT box

would destabilize the entire CCR4-NOT. Do the authors anticipate that CNOT3 lacking the NOT box is still able to associate with ribosomes but unable to module translation?

To further investigating whether the NOT box-truncated CNOT3 can associate with the translation machinery, we used the CNOT3 FL and NOT box truncated overexpressing cell lines to perform PLA using Flag antibody paired with antibodies raised against puromycin, EEF1D and EEF1E to specifically examine CNOT3 association between the elongating peptides as well as EEF1D and EEF1E. Despite some non-specific signals with Flag antibody in the empty vector (EV) control cells, we observed that overexpression of CNOT3-FL significantly enhanced the PLA signals. On the other hand, PLA signals detected in cells carrying the NOT box truncated CNOT3 were similar to those in EV control. The results indicate that deletion of the NOT box domain compromised association of CNOT3 with the translation machinery. The data shown below is also now included in the manuscript as extended data figure 6D-I.

Figure 7: Their model implies that mRNA is under 3 prime to 5 prime decay at the same time as undergoing increased translation, but this is not supported by their data.

We thank the reviewer for the comment. We have revised the model accordingly.

Syntax and readability issues:

Line 341: 'for examples'

We edited the text.

Line 373: 'among the translationally downregulate 'transcript'

We edited the text.

There is a mixture of cMyc and c-Myc across texts and figures.

We edited the text.

The word “diseases” in line 91 should not be plural

We edited the text.

The sentence in lines 495-497 is not grammatically correct and instead should read, “a pattern emerged showing genes that/which exhibited low G/C and high A/T content and enriched in some rare codons (i.e., lysine) to be preferentially promoted by CNOT3.”

We edited the text.

REVIEWERS' COMMENTS

Reviewer #1 (Remarks to the Author):

Review for the manuscript NCOMMS-23-41711A by Dr. Vu and Co-Authors Titled "Translation Efficiency Driven by the CNOT3 Subunit of the CCR4-NOT Complex Promotes Leukemogenesis"

The authors have addressed most of my previous concerns. I support the publication of manuscript in Nature Communications.

Reviewer #2 (Remarks to the Author):

The authors have sufficiently answered this reviewer's critiques.

Answer to question, bottom page 10: are proteome and transcriptome switched in Venn diagrams?

Reviewer #3 (Remarks to the Author):

The reviewers provided a thorough and diligent response to my critiques, and additional data has strengthened this manuscript further. I have no additional comments.

Eugene Valkov, NCI/NIH, USA.

REVIEWERS' COMMENTS

Reviewer #1 (Remarks to the Author):

Review for the manuscript NCOMMS-23-41711A by Dr. Vu and Co-Authors Titled "Translation Efficiency Driven by the CNOT3 Subunit of the CCR4-NOT Complex Promotes Leukemogenesis"

The authors have addressed most of my previous concerns. I support the publication of manuscript in Nature Communications.

We sincerely thank the reviewer for supporting the publication of our work.

Reviewer #2 (Remarks to the Author):

The authors have sufficiently answered this reviewer's critiques.

Answer to question, bottom page 10: are proteome and transcriptome switched in Venn diagrams?

We sincerely thank the reviewer for supporting the publication of our work.

We are thankful that the reviewer noticed the switching of labels in the Venn diagrams. The correct Venn diagrams is as below

Reviewer #3 (Remarks to the Author):

The reviewers provided a thorough and diligent response to my critiques, and additional data has strengthened this manuscript further. I have no additional comments.

Eugene Valkov, NCI/NIH, USA.

We sincerely thank the reviewer for recognizing our efforts and supporting the publication of our work.